# *HairFree:* Compositional 2D Head Prior for Text-Driven 360° Bald Texture Synthesis

**Mirela Ostrek**[1,2]    **Michael J. Black**[1]    **Justus Thies**[1,2]

[1]Max Planck Institute for Intelligent Systems [2]Technical University of Darmstadt

## Abstract

Synthesizing high-quality 3D head textures is crucial for gaming, virtual reality, and digital humans. Achieving seamless 360° textures typically requires expensive multi-view datasets with precise tracking. However, traditional methods struggle without back-view data or precise geometry, especially for human heads, where even minor inconsistencies disrupt realism. We introduce *HairFree*, an unsupervised texturing framework guided by textual descriptions and 2D diffusion priors, producing high-consistency 360° bald head textures—including non-human skin with fine details—***without any texture, back-view, bald, non-human, or synthetic training data***. We fine-tune a diffusion prior on a dataset of mostly frontal faces, conditioned on predicted 3D head geometry and face parsing. During inference, *HairFree* uses precise skin masks and 3D FLAME geometry as input conditioning, ensuring high 3D consistency and alignment. We synthesize the full 360° texture by first generating a frontal RGB image aligned to the 3D FLAME pose and mapping it to UV space. As the virtual camera moves, we inpaint and merge missing regions. A built-in semantic prior enables precise region separation—particularly for isolating and removing hair—allowing seamless integration with various assets like customizable 3D hair, eyeglasses, jewelry, etc. We evaluate *HairFree* quantitatively and qualitatively, demonstrating its superiority over state-of-the-art 3D head avatar generation methods. https://hairfree.is.tue.mpg.de/

## 1 Introduction

Generating realistic and consistent textured 3D head avatars is essential for applications in gaming, virtual reality, and digital human modeling. Achieving 360° appearance consistency in head texturing is a persistent challenge, especially, because scalp visibility varies dramatically with dynamic hair animations and different hairline shapes (e.g., straight, rounded, widow's peak, M- or V-shaped, receding), see Figure 1. Existing methods [15, 16, 17, 14, 28, 30, 40, 31, 34, 64, 2, 70] either bake hair (and its specific hairline) directly into the texture—leading to visible artifacts when swapping to any other hairstyle—or limit textures to the facial region, neglecting the full scalp. Moreover, other approaches rely on large-scale multi-view datasets of human heads with precise tracking, making them resource-intensive and less flexible.

This lack of flexibility undermines true compositionality, where any 3D hairstyle or other assets (hats, helmets, eyeglasses, jewelry) can be seamlessly added, swapped, or animated without visible seams or mismatches. For practical applications, a fully disentangled, hair-free scalp and face texture is crucial, serving as a neutral base for compositional 3D layering. In this paper, we introduce *HairFree*, an unsupervised generative texturing framework that generates high-quality, 360° head textures, providing a fully bald, neutral base for 3D asset integration. Unlike existing methods, *HairFree* uses a diffusion-based inpainting approach guided by textual descriptions and produces consistent, detailed head textures without relying on any texture, back-view, bald, non-human, or synthetic training

39th Conference on Neural Information Processing Systems (NeurIPS 2025).

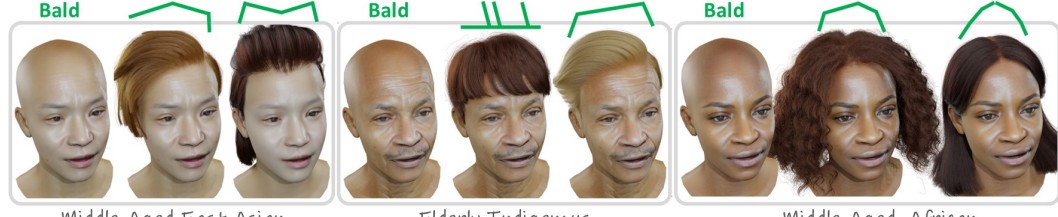

Middle-Aged East Asian      Elderly Indigenous      Middle-Aged African

Figure 1: *HairFree* is a hybrid 2D/3D neural rendering method that synthesizes diverse, high-fidelity 360° bald human head textures given *a 3D head mesh* and *a textual description*. The generated bald textures (columns 1, 4, 7) seamlessly integrate into classical graphics pipelines, allowing compatibility with any 3D hairstyle—regardless of hairline type (in green)—in a fully compositional manner.

data. Our method is based on a compositional 2D human head prior, trained on 100K mostly frontal images of faces, conditioned on RGB background, 3D head geometry, and partial facial semantics. The 3D head geometry and facial semantics are estimated using off-the-shelf methods (Spectre [12], FPM [68]).

At inference time, *HairFree* begins by generating a high-quality frontal view image of a face, conditioned on accurate 3D FLAME geometry [35] and precise skin masks. This initial image is projected onto the 3D FLAME texture space, creating a base texture. As the virtual camera moves around the head, a progressive inpainting process is applied—rendering the visible regions and filling in missing areas using the diffusion model in image space. These completed regions are then re-mapped to texture space, maintaining texture consistency. The entire process is guided by the same 3D-consistent FLAME geometry and skin masks, ensuring precise alignment across all views. The semantic conditioning allows for skin-hair separation, effectively isolating and removing hair, producing a clean bald texture. The final result is a complete 360° bald head texture, fully compatible with various 3D assets, such as strand-based hairstyles (Figure 1).

Our results demonstrate that *HairFree* generates high-quality, consistent, and detailed 360° textures for fully bald heads, including both human skin (Figure 4) and non-human skin textures (Figure 5). This opens up new possibilities for creating customizable 3D head avatars using textual descriptions, without the need for extensive multi-view datasets or supervised training. Finally, we evaluate *HairFree* both quantitatively and qualitatively, showing its superiority over state-of-the-art methods.

**In summary**, we make the following contributions:

- *Compositional 2D head prior:* a diffusion model that generates high-quality, photorealistic face images, conditioned on 3D geometry, facial semantics, and text prompts, enabling precise separation of hair and skin regions.

- *360° bald head texturing pipeline:* a robust, unsupervised texturing method that synthesizes consistent, high-quality 360° bald head textures from text prompts. The pipeline leverages our compositional 2D head prior in combination with a generic inpainting prior. This method integrates image-space inpainting with texture mapping onto a FLAME-based head mesh in UV space. It generates full head textures without relying on any texture, back-view, bald, non-human, or synthetic data, and generalizes to both human and non-human skin types.

- *A dataset* of 1,000 generated, high-quality, photorealistic human head textures, providing a diverse and scalable resource for realistic avatar and face model development.

## 2   Related Work

Related work spans three areas: 2D image synthesis, mesh texturing, and 3D character generation. While GAN- and diffusion-based models excel at face image creation and 3DMM or reconstruction methods yield plausible head textures, none offer a compositional 2D prior that cleanly separates hair and skin for full-head texturing on arbitrary meshes. Below, we review key advances in each area and show how our approach fills this gap.

**2D Image Synthesis:** Generative Adversarial Networks (GANs) [19] and their StyleGAN successors [25, 23, 24, 3] have set the standard for photorealistic image generation across objects and human faces [61, 62, 32, 56], with StyleAvatar extending StyleGAN to texture maps for 3DMM-based avatars [65, 63]. More recently, diffusion models [48, 52, 51, 1, 55, 45, 44] trained on LAION-5B [57] surpass GANs in quality and diversity, powering robust text-to-image synthesis [54] and supporting fine-grained control through arbitrary image-conditioning—landmarks, segmentation masks, depth maps, or rendered geometry—via ControlNet [71]. We fine-tune Stable Diffusion with ControlNet conditioning to provide precise, identity-preserving guidance for our avatar texturing pipeline.

**Texturing and Face Textures:** 3D morphable models (3DMMs) [10], such as the Basel Face Model [47], use PCA on textured scans to represent facial geometry and texture, becoming standard for face tracking [72] and neural rendering techniques like NeRF [43] and Gaussian splatting [27]. However, their texture spaces lack diversity due to limited 3D data used to create it. Methods like FlameTex [11], Slossberg et al. [59], Gecer et al.[16, 17], and CLIPFace [2] expand texture variety using in-the-wild images. CLIPFace uses the FLAME model [35] with a StyleGAN-like architecture for high-quality textures. DreamFace [70] uses CLIP-based selection for coarse geometry, then refines details with Score Distillation Sampling (SDS), combining generic and texture latent diffusion models to generate diverse, high-quality frontal textures of the facial region. FitMe [29] (GAN inversion), Relightify [46] (diffusion), and Luo et al. [40] (StyleGAN) reconstruct photorealistic facial textures directly from images but also cover only faces, not full heads, same as UV-IDM [33]. Most of these methods bake at least some hair into the texture, preventing compositional layering with separate hair assets.

**3D Character Generation:** Several recent works generate 3D textures and geometry via "generation by reconstruction," synthesizing multiple 2D views and then lifting them into 3D [37, 49, 4, 42, 53, 13, 7]. General-purpose methods like TEXTure [53], Text2Tex [6], and SceneTex [5] use diffusion-based priors in a generate-then-refine pipeline to texture arbitrary objects and scenes. Human-specific approaches include TADA [36] and HumanNorm [21], which recover full body assets via DMTet [58] and SDS optimization [49], and FaceLift [41], which directly predicts multi-view images with a latent diffusion model before reconstructing with Gaussian splatting. TECA [69] further introduces mesh-volumetric disentanglement of skin and hair under an SDS loss. Arc2Avatar [18] produces high-quality 3D heads from single images by leveraging a human face foundation model and full 3DMM integration for superior realism and identity preservation. None of these methods are designed to generate specifically bald textures or fully disentangle hair. In contrast, our compositional 2D diffusion prior disentangles hair from skin in a single-stage process, yielding fully editable, high-fidelity 360° head textures.

## 3 Preliminaries: Diffusion Models

Our 2D human head prior is a latent diffusion model (LDM) [54], fine-tuned to transform rendered head meshes, partial face parsing masks, and RGB backgrounds into photorealistic human heads.

**Denoising Diffusion Probabilistic Model (DDPM):** In the DDPM framework [20], noise-corrupted samples are progressively denoised over $T$ timesteps. The forward process adds Gaussian noise $\mathcal{N}(\mathbf{0}, \mathbf{I})$ in $T$ steps using a variance schedule $\{\beta_t\}$:

$$q(\mathbf{x}_t|\mathbf{x}_{t-1}) = \mathcal{N}(\mathbf{x}_t; \sqrt{1-\beta_t}\mathbf{x}_{t-1}, \beta_t\mathbf{I}). \tag{1}$$

The reverse denoising process, parameterized by $\epsilon_\theta$, approximates $p_\theta(\mathbf{x}_{t-1}|\mathbf{x}_t)$:

$$p_\theta(\mathbf{x}_{t-1}|\mathbf{x}_t) = \mathcal{N}(\mathbf{x}_{t-1}; \boldsymbol{\mu}_\theta(\mathbf{x}_t, t), \boldsymbol{\Sigma}_\theta(\mathbf{x}_t, t)), \tag{2}$$

with an objective function:

$$\mathcal{L}_{\text{DDPM}} = \mathbb{E}_{\mathbf{x}_0, \epsilon \sim \mathcal{N}(0,1), t}\left[\|\epsilon - \epsilon_\theta(\mathbf{x}_t, t)\|_2^2\right]. \tag{3}$$

**Latent Diffusion Model (LDM):** Operating in the latent space, LDMs [54] use a pre-trained VAE to map images to latent codes, reformulating the objective as:

$$\mathcal{L}_{\text{LDM}} = \mathbb{E}_{\mathcal{E}(\mathbf{x}), \epsilon \sim \mathcal{N}(0,1), t}\left[\|\epsilon - \epsilon_\theta(\mathbf{z}_t, t)\|_2^2\right], \tag{4}$$

where $\mathbf{z}_t$ is the latent code at timestep $t$.

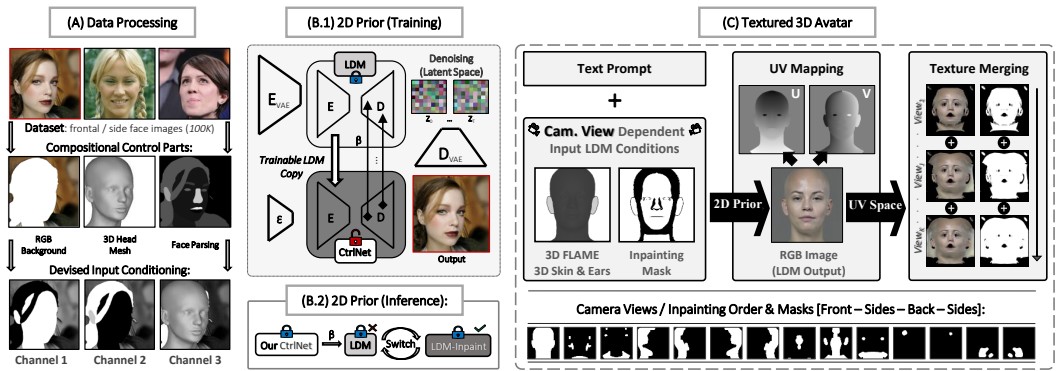

Figure 2: **System Overview:** (A) *Data Processing:* Estimate face parsing, 3D head mesh, and remove the foreground to form compositional inputs. (B.1) *Training Prior:* Fine-tune an LDM via ControlNet using these inputs and a generic "face" prompt. (B.2) *Inference Prior:* Swap in a generic LDM-Inpainting prior. (C) *Texture Generation:* Generate a frontal view, map to UV space, iteratively render "seen" and inpaint "unseen" regions while moving the camera, building a full 360° texture.

## 4  Method

Figure 2 outlines our system. We first train a compositional 2D head prior via latent diffusion on predominantly frontal face images, conditioned on rendered FLAME meshes, face parsing masks, and backgrounds. At inference, we use the FLAME model to condition the generation of a frontal view, project it into UV space, and then iteratively fill in unseen texture regions. The following sections describe (1) our diffusion prior and its training and (2) the progressive texturing process.

### 4.1  Compositional 2D Human Head Prior

We fine-tune a Latent Diffusion Model (LDM) [54] using ControlNet [71] to generate high-quality images conditioned on specific input features. ControlNet operates as a *trainable copy* of the base diffusion model (Stable Diffusion 2.1 [54]), which is kept frozen in *locked mode* during training. This ensures stable training while ControlNet learns to map conditions from a dataset of 100K images (FFHQ [26] and CELEB-A-HQ [22]).

We condition our model using a set of inputs, collectively denoted as $\mathbf{C}$, which include skin masks ($\mathbf{c}_{\text{skin}}$), hair masks ($\mathbf{c}_{\text{hair}}$), ears and accessory masks ($\mathbf{c}_{\text{ears}}$), a 3D head mesh ($\mathbf{c}_{\text{mesh}}$) rendered in 2D, and background information ($\mathbf{c}_{\text{background}}$). These conditions are combined as follows:

$$\mathbf{C} = \{\mathbf{c}_{\text{skin}}, \mathbf{c}_{\text{hair}}, \mathbf{c}_{\text{ears}}, \mathbf{c}_{\text{mesh}}, \mathbf{c}_{\text{background}}\}. \tag{5}$$

The training objective is defined as:

$$\mathcal{L}_{\text{COND}} = \mathbb{E}_{\mathbf{z}_0, \mathbf{t}, \mathbf{C}, \epsilon \sim \mathcal{N}(0,1)} \left[ \|\epsilon - \epsilon_\theta(\mathbf{z}_t, \mathbf{t}, \mathbf{C})\|_2^2 \right]. \tag{6}$$

To further enable text-based conditioning, we employ a CLIP text encoder, transforming prompts into embeddings that guide image generation via cross-attention layers. Classifier-free guidance (CFG) is applied to balance text adherence and image quality, adjusting the model's prediction as:

$$\epsilon_\theta(\mathbf{z}_t, \mathbf{C}) = (1 + \omega) \cdot \epsilon_\theta(\mathbf{z}_t, \mathbf{C}) - \omega \cdot \epsilon_\theta(\mathbf{z}_t), \tag{7}$$

where $\omega$ controls the guidance strength.

**Guided Inpainting:**   At inference, we replace the locked LDM with an inpainting variant, allowing for guided completion of missing texture regions. We introduce an inpainting mask ($\mathbf{c}_{\text{mask}}$) specifying areas to be filled, while maintaining coherence with the existing texture using the same conditioning set $\mathbf{C}$. The inpainting objective is defined as:

$$\mathcal{L}_{\text{INP}} = \mathbb{E}_{\mathbf{z}_0, \mathbf{t}, \mathbf{C}, \mathbf{c}_{\text{mask}}, \epsilon \sim \mathcal{N}(0,1)} \left[ \|\epsilon - \epsilon_\theta(\mathbf{z}_t, \mathbf{t}, \mathbf{C}, \mathbf{c}_{\text{mask}})\|_2^2 \right]. \tag{8}$$

**Training Details:**   Our prior is trained for $\sim 1500$ GPU hours on an NVIDIA H100 (see [54, 71]).

## 4.2 3D Texturing Pipeline

First, we render skin and ear masks aligned to the 3D FLAME model [35] from multiple viewpoints and use them to condition our 2D diffusion prior, producing consistent pixel outputs. Next, these pixel colors are projected onto the mesh's UV atlas in an iterative process: visibility checks ensure only previously untextured regions are filled, while morphological erosion and bilinear interpolation refine boundaries and smooth transitions. Together, these steps yield a high-quality 360° head texture.

**Rendering 3D-Consistent Input Controls:** At test time, we generate 3D-consistent conditioning signals using the FLAME model. Specifically, we extract skin and ear maps aligned with the FLAME mesh, while omitting features like hair, earrings, and eyeglasses to ensure clean, bald head generation. The background is set to a uniform color to minimize artifacts, keeping the model's focus on the head region. The FLAME model, a 3D Morphable Model (3DMM), parameterizes human head shapes and expressions through a low-dimensional latent space. It outputs a 3D head mesh $\mathbf{M} = f_{\text{FLAME}}(\alpha, \delta, \theta)$, where $\mathbf{M}$ is represented by vertices $\{\mathbf{v}_i \in \mathbb{R}^3\}_{i=1}^N$ and a fixed topology of faces $\mathbf{F}$. These vertices are controlled by shape parameters $\alpha$, expression parameters $\delta$, and pose parameters $\theta$, enabling precise manipulation of head structure and expressions.

To capture a complete 360° representation, we render the mesh from 14 viewpoints by rotating a virtual camera around the head. Each vertex $\mathbf{v}_i = (x_i, y_i, z_i)^\top$ is projected onto the 2D image plane using a perspective transformation, where the camera intrinsics $\mathbf{K}$ and extrinsics $\mathbf{R}, \mathbf{t}$ define the projection as $\mathbf{p}_i = \pi(\mathbf{K}(\mathbf{R}\mathbf{v}_i + \mathbf{t}))$. The perspective division is given by $\pi(\mathbf{x}) = \left(\frac{x}{z}, \frac{y}{z}\right)$ for a 3D point $\mathbf{x} = (x, y, z)^\top$.

These rendered meshes and accurate skin/ear segmentation masks serve as the conditioning signals for our compositional 2D image prior.

**Progressive UV Mapping:** Guided by our 3D-consistent input controls, the generated images of the 2D prior align with the FLAME mesh. Each vertex $\mathbf{v}_i$ on the 3D mesh has a corresponding UV coordinate $(u_i, v_i)$, allowing us to map surface points to a 2D texture space. For each view, visible pixels on the 2D render are mapped to UV coordinates $(u, v)$ based on the surface-to-UV correspondence. Each pixel $(x, y)$ in the rendered frame has an RGB color value $\mathbf{col}(x, y) = [r, g, b]$, which we splat to the texture space at the corresponding $(u, v)$ locations.

Instead of computing the images of all views at once, we iteratively render the images using already existing texture parts. To improve texture quality and avoid blending artifacts near boundaries, we apply a morphological erosion operation to the existing UV texture mask before accumulating new color information. Specifically, for a pixel $(x, y)$, we compute its corresponding UV coordinates $(u, v)$ using precomputed channels:

$$
\begin{aligned}
u &= \lfloor u_{\text{channel}}(y, x) \cdot R \rfloor, \\
v &= \lfloor v_{\text{channel}}(y, x) \cdot R \rfloor,
\end{aligned}
\tag{9}
$$

where $u_{\text{channel}}$ and $v_{\text{channel}}$ are UV maps providing $(u, v)$ coordinates for each pixel and $R$ denotes the image resolution. These $(u, v)$ values are rounded down to the nearest integer for indexing into the texture map.

We conditionally update the UV texture based on visibility checks, ensuring that only new visible regions accumulate:

$$
\mathbf{T}(u, v) = \begin{cases} \mathbf{col}(x, y) & \text{if visible at } (u, v), \\ \mathbf{T}(u, v) & \text{otherwise.} \end{cases}
\tag{10}
$$

For each pixel $(x, y)$ with color $\mathbf{col}(x, y) = [r, g, b]$ in the $512 \times 512$ image space, we compute its UV coordinates on the $1024 \times 1024$ atlas and define $u_0 = \lfloor u \rfloor$, $u_1 = \lceil u \rceil$, $v_0 = \lfloor v \rfloor$, and $v_1 = \lceil v \rceil$. We then "splat" $\mathbf{col}(x, y)$ into each of the four texels $(u_k, v_\ell) \in \{(u_0, v_0), (u_1, v_0), (u_0, v_1), (u_1, v_1)\}$ by setting:

$$
\mathbf{T}(u_k, v_\ell) = \mathbf{col}(x, y) \quad \text{if that texel is not yet filled.}
\tag{11}
$$

This effectively closes small gaps and holes in the accumulated texture. Applying this process across all pixels and viewpoints yields a full 360° UV texture.

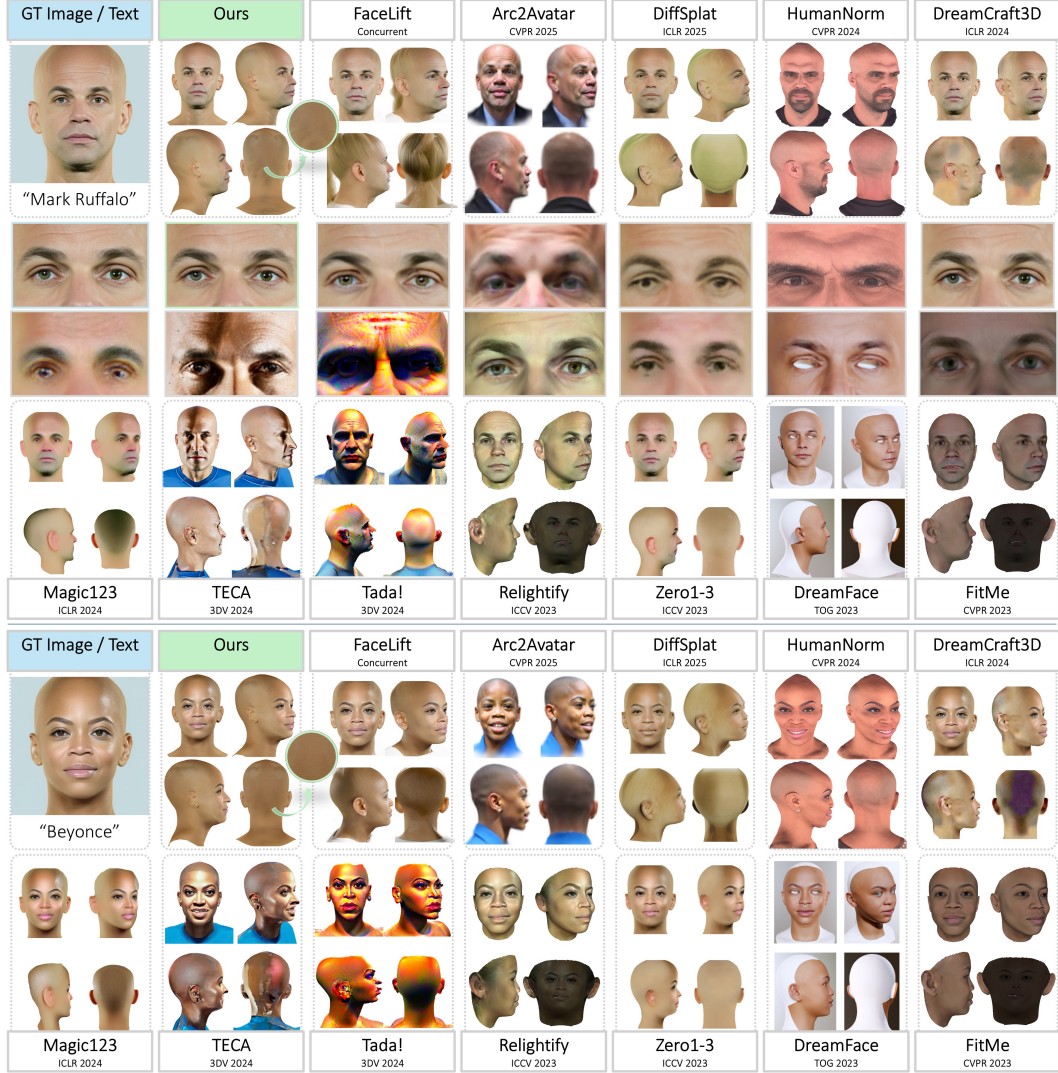

Figure 3: **Qualitative Comparison with State-of-the-Art:** We compare *HairFree* against recent 3D avatar techniques—FaceLift [41], Arc2Avatar [18], DiffSplat [38], HumanNorm [21], Dream-Craft3D [60], Magic123 [50], TECA [69], TADA! [36], Relightify! [46], Zero123-XL [39], Dream-Face [70], and FitMe [29]—on "Mark Ruffalo" and "Beyonce" with explicit bald-head constraints. *HairFree* delivers the most accurate 360° head shapes, realistic textures, and uniform lighting.

## 5 Experiments

We evaluate our proposed method in two critical aspects: (1) the compositional 2D human head prior and (2) the texture generation pipeline. Our 2D head prior offers precise, region-specific control over facial components, including skin, hair, and ears. The texture generation pipeline is assessed based on its ability to produce photorealistic, high-quality human head textures, maintaining diversity in ethnicity, age, and style, ensuring accurate texture synthesis throughout the full 360° range.

### 5.1 2D Human Head Prior

**Comparison with State-of-the-Art 2D Bald Proxy Methods:** To qualitatively evaluate the 2D human head prior, we compare it as a bald proxy method with the following approaches: (i) diffusion-based generic inpainting (LDM [54]), (ii) GAN-based 2D bald proxy estimation (HairMapper [67]), and (iii) the state-of-the-art hair editing method HairCLIPv2 [66]. The results highlight the ability of

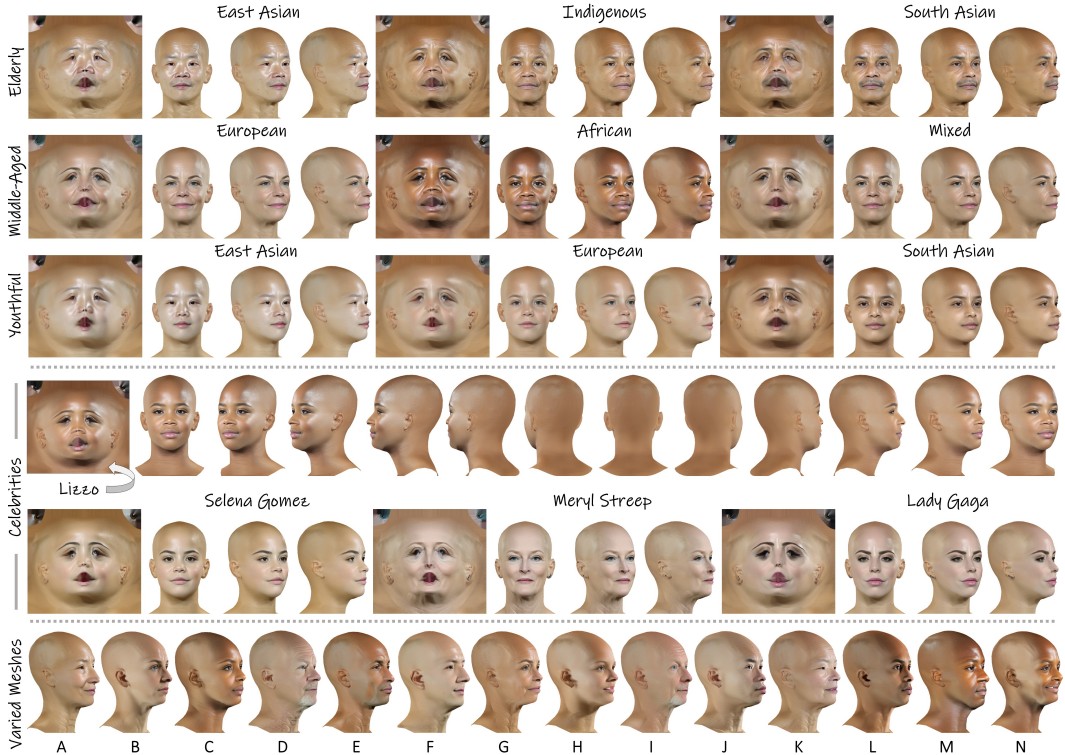

Figure 4: **Photorealistic Rendering Results.** Without using any captioned text prompts during training, our method accurately follows text prompts at test time. *(A) Demographics Attributes:* Rendered textures demonstrating variations in age and demographics across head avatars. *(B) Celebrity Renderings:* Multi-view renderings and textures of famous celebrities. *(C) Varied Meshes and Nationalities:* Showing additional variety in nationality-based text prompts applied to different meshes. Corresponding text prompts, additional results and camera views are available in Appendix A.

our approach to preserve more accurate head shapes and poses while effectively removing hair. The results are shown in Appendix A.

**Quantitative Evaluation - ControlNet Strength (CS):**   We evaluate our model using FID, KID, LPIPS, and PSNR, across different ControlNet strengths (CS). By reducing CS, we can generate more diverse examples, enabling the synthesis of novel head views that do not exist in the training data, see Table 2 and Figure 6 (C).

**Ablation - Inpainting (RGB):**   We compare the performance of our approach with and without inpainting. As shown in Figure 6 (A), the model without inpainting is limited to generating accurate frontal views, while using gradual inpainting from front to back enables the synthesis of a broader range of views, including extreme side, back, and top perspectives (not present in the training data).

### 5.2   3D Textured Avatar Synthesis

**Comparison with State-of-the-Art 3D Methods:**   We conduct a qualitative comparison against recent state-of-the-art 3D avatar generation methods. Zero123-XL [39] and Magic123 [50] rely on 2D diffusion-based priors using Score Distillation Sampling [49], which often leads to oversaturation, low detail preservation, and inconsistent geometry across views. Tada![36], which is specialized for humans, is also SDS-based and struggles with oversaturation, while HumanNorm[21] introduces unnatural reddish skin tones. DreamCraft3D [60] and DiffSplat [38] leverage 3D Gaussian splatting, which improves efficiency but tends to produce low-frequency details and blurry textures. TECA [69] generates clothing in addition to the head. FaceLift [41] produces relatively accurate head shapes but may hallucinate hair instead of adhering to the bald constraint, and like many other methods, it suffers from extreme view-specific illumination artifacts (e.g., shadows). Arc2Avatar [18], Relightify [46],

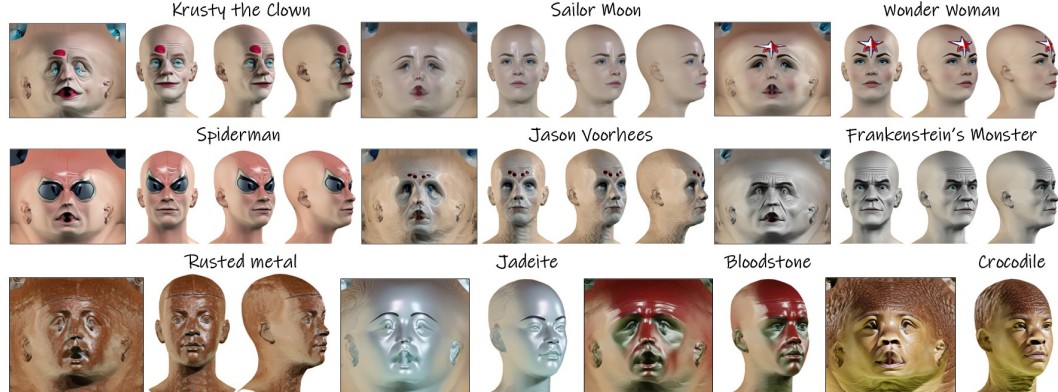

Figure 5: **Generalization Properties.** Our method enables blending between photorealistic and stylized renderings through the classifier-free guidance (CSG) and control strength (CS) parameters. Despite being fine-tuned only on realistic human faces, it generalizes strongly to *diverse, non-uniform* bald textures, enabling appearances beyond natural human skin. **CSG:** Impacts the alignment with the text. **CS:** Lower values relax the model's adherence to our prior, allowing for greater style diversity (see row 3). Additional results are available in Appendix A.

| Method | Runtime | Stages | Quality |
|---|---|---|---|
| FitMe | minutes | 2 | Mid |
| DreamFace | minutes | 2 | Mid |
| Zero1-3 | minutes | 1 | Low |
| Relightify | minutes | 2 | Mid |
| Tada! | 1 hour | 1 | Low |
| TECA | minutes | 1 | Low |
| Magic123 | 1 hour | 2 | Low |
| DreamCraft3D | 3 hours | 4 | High |
| HumanNorm | 1 hour | 3 | Mid |
| DiffSplat | seconds | 1 | Mid |
| Arc2Avatar | 1-2 hours | 2-3 | Mid |
| FaceLift | seconds | 1 | High |
| HairFree (Our) | minutes | 1 | **High** |

Table 1: **Quantitative Comparison with State-of-the-Art:** Runtime, stages, quality.

| CS | PSNR ↑ | MSE ↓ | LPIPS ↓ | FID ↓ | KID ↓ |
|---|---|---|---|---|---|
| **1.0** | **18.38** | **0.018** | **0.24** | **6.6** | **0.0027** |
| 0.75 | 17.14 | 0.023 | 0.29 | 11.5 | 0.0056 |
| 0.5 | 15.05 | 0.035 | 0.39 | 27.9 | 0.0167 |
| 0.25 | 12.53 | 0.060 | 0.53 | 78.9 | 0.0518 |
| 0 | 10.27 | 0.097 | 0.7268 | 384 | 0.4706 |

Table 2: **Quantitative Analysis/Ablation:** We vary ControlNet Strength (CS) from 0 to 1 and measure image similarity over 10K samples. At CS = 1.0, outputs closely match the input; as CS decreases, diversity increases, enabling the synthesis of bald, back-view, and extreme side-views.

DreamFace [70], and FitMe [29] all struggle to generate a complete, bald scalp. Arc2Avatar produces a full 360° avatar but suffers from entanglement issues, with visible clothing and a non-bald scalp showing a hairline instead of clear skin. Relightify and FitMe both reconstruct texture and geometry together rather than focusing on texturing, limiting their ability to produce consistent high-quality results. Neither method effectively addresses hair removal, and their outputs are of medium quality. DreamFace is further limited to generating only the face region without the scalp, and has limited skin diversity. Most of the baselines fail to enforce baldness, introducing unwanted hair despite explicit constraints. Additional quantitative analysis with respect to runtime, method complexity, and image quality is given in Table 1. Our approach achieves the highest fidelity 360° textures, setting a new benchmark for textured 3D avatar quality (Figure 3).

**Qualitative Evaluation:** We demonstrate the diversity and realism of our textures across various text prompts, including photorealistic, celebrity, and stylized outputs (Figures 4 and 5). Additional results covering fantasy, artistic styles, animal faces, and material synthesis are in Appendix A.

**Ablation - Classifier-Free Guidance (CSG):** In Figure 6 (B), we demonstrate how varying CSG affects the alignment of the output with text. Higher CSG values lead to outputs more aligned with the text prompt, while lower values relax this alignment and result in a photorealistic appearance.

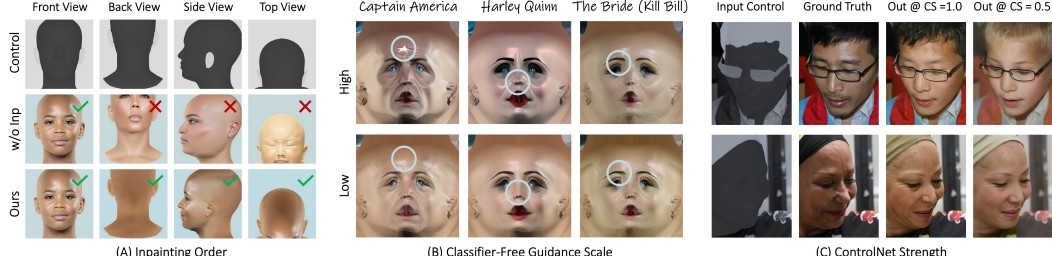

Figure 6: **Ablation Studies:** (A) Without our progressive inpainting, only frontal views are accurate. (B) Higher CSG values improve adherence to text prompts; lower values produce a more natural look. (C) Reducing ControlNet strength gradually weakens alignment with the input conditioning signal.

**User Study:** We conducted a user study for Person 1 (87 users) and Person 2 (94 users), asking to select the top 1 method in terms of avatar plausibility. Each user was shown one frontal, two side, and one back view of avatars generated by 13 different methods (Figure 3, Appendix A). For Person 1, *HairFree* led with 40.23% of participants, followed by *Dream-Craft3D* at 18.39%, *FaceLift* and *Arc2Avatar* tied at 8.05% each, and *DiffSplat* with 6.90%. For Person 2, *HairFree* led with 39.36%, followed by *DreamCraft3D* at 14.89%, *FaceLift* at 9.57%, and *Arc2Avatar* and *DiffSplat* tied at 7.45% each. These results confirm the superiority of our method (Figure 7).

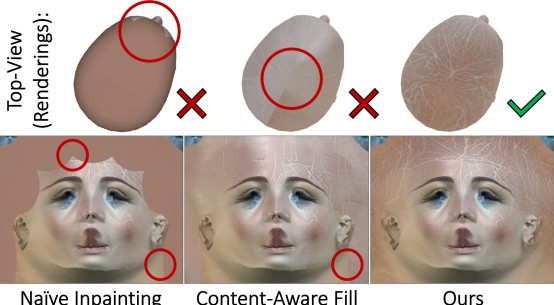

Figure 7: **User Study:** Distribution of user preferences among the selected methods for two individuals, highlighting the preference for our method.

**Comparison with Inpainting Methods:** Figure 8 compares three approaches for inpainting in the UV space: (1) a naive method where all non-frontal views are filled with a uniform color, resulting in visible seams, (2) Content-Aware Fill (CAF)-based inpainting in UV space, which also produces seams, particularly at the back, and (3) our method, which performs inpainting directly in image space. Unlike the other methods, our approach seamlessly preserves intricate patterns without any visible seams in the UV space, even for intricate non-human skin textures.

Figure 8: **Comparisons:** Inpainting in UV Space.

**Limitations & Societal Impact:** Our method is sensitive to lighting inconsistencies—addressable via intrinsic decomposition (e.g., IntrinsicAnything [8]) for relightable textures. Text prompts inherit any ethnicity or appearance biases from the pre-trained diffusion prior. High-fidelity textures could enable identity spoofing or deepfakes, so responsible, careful use is crucial.

## 6 Conclusion

We introduced *HairFree*, a diffusion-based framework that generates realistic, 3D-consistent bald head textures by conditioning a large latent diffusion model on face parsing maps, 3D meshes, and background cues. During inference, guided inpainting fills unseen regions as the camera moves, yielding seamless 360° textures. Our compositional 2D prior cleanly separates skin from hair, enabling flexible 3D layering of external assets—such as strand-based hairstyles—independent of hairline. Our evaluations demonstrate that *HairFree* delivers state-of-the-art fidelity and compositionality compared to existing 3D head avatar methods.

**Acknowledgements:** The authors thank Tsvetelina Alexiadis, Tomasz Niewiadomski, and Taylor Obersat for perceptual study; Yao Feng, Tingting Liao, Dimitrios Gerogiannis, Weijie Lyu, Alexandros Lattas, and Foivos Paraperas Papantoniou for help with the baselines; Peter Kulits, Yuliang Xiu, and all reviewers for their valuable feedback; and Benjamin Pellkofer for IT support. Justus Thies is supported by the DFG Excellence Strategy— EXC-3057 and the project is co-funded by the European Union (ERC, Lemo, 101162081). Views and opinions expressed are however those of the author(s) only and do not necessarily reflect those of the European Union or the European Research Council. Neither the European Union nor the granting authority can be held responsible for them.

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

# A    Appendix

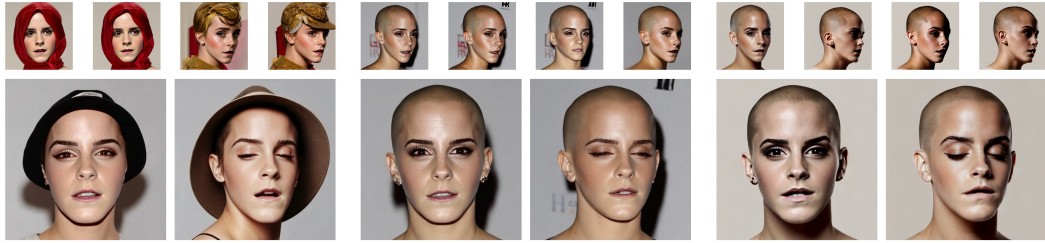

rendered mesh, hair mask               + accessories (except from earrings)               + background

Figure 9: Outputs from three models trained with progressively richer conditioning inputs: (left) rendered mesh with hair mask only; (middle) + accessories (excluding earrings); (right) + background. All examples are from an early training epoch and highlight the effect of each conditioning signal. The model gradually learns to associate additional inputs with their corresponding visual semantics. Note that earrings are not included in the conditioning, which causes them to appear or disappear inconsistently across examples. The final model, trained for longer, achieves more stable results and can fully remove hair during inference.

## A.1    Ablation Study on Semantic Conditioning Signals

The face and skin mask covers all facial regions including eyes, nose, lips, etc, teaching the model what skin and facial features look like (see "Devised Input Conditioning" in Figure 2). The hair mask is used during training only, helping the model learn hair locations so it can remove hair at inference (e.g., for bald heads). Ear masks are used during both training and inference; at inference, we use precise 3D ear masks from FLAME for accurate, mesh-aligned ear synthesis. Accessory masks allow the model to learn to recognize and remove the accessories by omitting the mask at test time. Note that during training time those semantic masks can not be derived from the 3D mesh. See Figure 9 and Figure 10 for an ablation study on semantic conditioning signals.

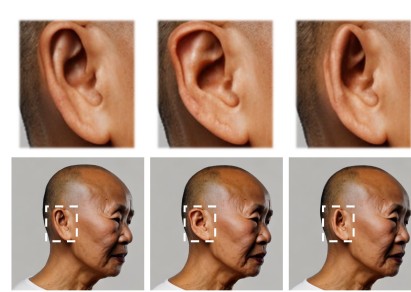

Figure 10: Examples showing inconsistent ear synthesis when ear masks are omitted. Each column shows a bald head generated from the same text prompt (e.g., "old Asian woman") with zoomed-in ear crops. Without explicit ear guidance, the model produces visible variations in ear shape, size, and placement across samples. This instability indicates that accurate, mesh-aligned ear-mask supervision is crucial for maintaining structural consistency during generation.

**Janus effect:**    The Janus effect arises because the ControlNet is trained mostly on frontal and side views. When directly conditioned on a back-view mesh and semantic mask, especially with the default ControlNet strength of 1.0, it tends to hallucinate a face, having never seen such inputs during training (see Figure 6 (A), Figure 11). Progressive inpainting resolves this by gradually completing the texture from front to back, so later views only need to fill in small missing regions. Lowering ControlNet strength during these steps prevents over-conditioning on unfamiliar views. Additionally, we use a single face and skin mask that combines all facial attributes (eyes, lips, nose, etc.) into one region. We avoid using separate masks for each part, as many of these features are small and prone to inaccuracies in the off-the-shelf face parsing model. See "Devised Input Conditioning" in Figure 2 for an example of this mask.

## A.2    Controlnet Strength Influence on Distribution Shift

The ControlNet strength has a major influence on shifting from one distribution to another, see Figure 12. Unfortunately, we don't have a large dataset of bald heads, where we could directly evaluate the generation of the backside of the head. However, we conducted an experiment on changing the distribution from human faces to cat faces (which is an extreme case of a distribution shift), where we could analyse the effects of the ControlNet Strength. Specifically, we applied our

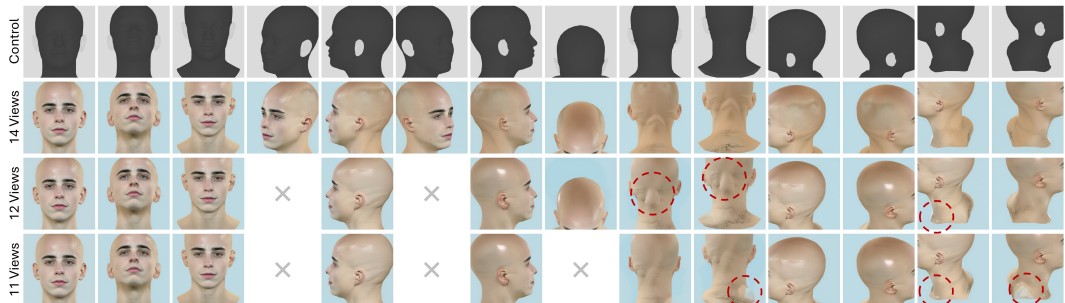

Figure 11: **Ablation on the number of conditioning views:** We evaluate the effect of reducing the number of input views used for texture generation. Row 1 shows the 14 control views. Row 2 shows our full setting with 14 views. Rows 3 and 4 show results with 12 and 11 views, respectively. Red highlights indicate artifacts that emerge as view coverage decreases.

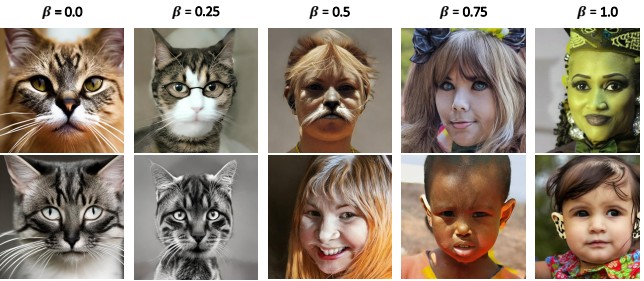

Figure 12: Visualization of the distribution shift between the cat and the human face distribution by controlling the ControlNet strength.

| $\beta$ | FID | KID |
|---|---|---|
| 1.0 | 231.32 | 0.2074 |
| 0.75 | 228.46 | 0.2035 |
| 0.5 | 212.10 | 0.1910 |
| 0.25 | 39.88 | 0.0237 |
| 0 | 81.93 | 0.0347 |

Table 3: Quantiative Controlnet strength influence (controlled with $\beta$) on distribution shift.

method to the AFHQ-Cat dataset from StarGAN v2 [9], a commonly used dataset from a significantly different domain. We measured FID and KID scores at various values, see Table 3. For higher values of ControlNet strength (e.g., $\beta >= 0.5$), the model remains biased toward synthesizing human faces. This prevents effective generation of samples from the cat distribution, resulting in high FID/KID scores and poor visual alignment with the AFHQ domain. At $\beta = 0.25$, the ControlNet guidance is reduced enough to allow the diffusion prior to generate realistic cat faces, while still retaining enough conditioning to preserve global head structure. This balance aligns well with the AFHQ-Cat distribution, as reflected in the significantly improved scores. At $\beta = 0$, although the model continues to produce cat faces, the samples are often zoomed-in facial crops rather than full cat heads, deviating from the AFHQ distribution. In contrast, at $\beta = 0.25$, structural guidance helps preserve head framing consistent with the dataset.

### A.3 Fixed & Varied Meshes with & without Hair

Figures 13 and 14 illustrate texture maps rendered on head models from three different views, highlighting the consistency of our method in maintaining photorealistic details. The examples span a diverse range of ethnicities and age groups, emphasizing the versatility of our approach in capturing the nuanced characteristics of human faces.

To explore the expressive capabilities of our method, Figures 15 and 16 contain examples generated from text prompts with "fantasy" elements. The rendered textures demonstrate the ability of our method to produce imaginative and stylistic results, maintaining consistency and coherence across different views.

To analyze the ability of our method to replicate artistic styles, Figures 17, and 18 present generations inspired by famous paintings and painting techniques. These results highlight how our approach captures distinct brushwork, color schemes, and compositional elements, preserving the essence of each referenced style while maintaining structural coherence.

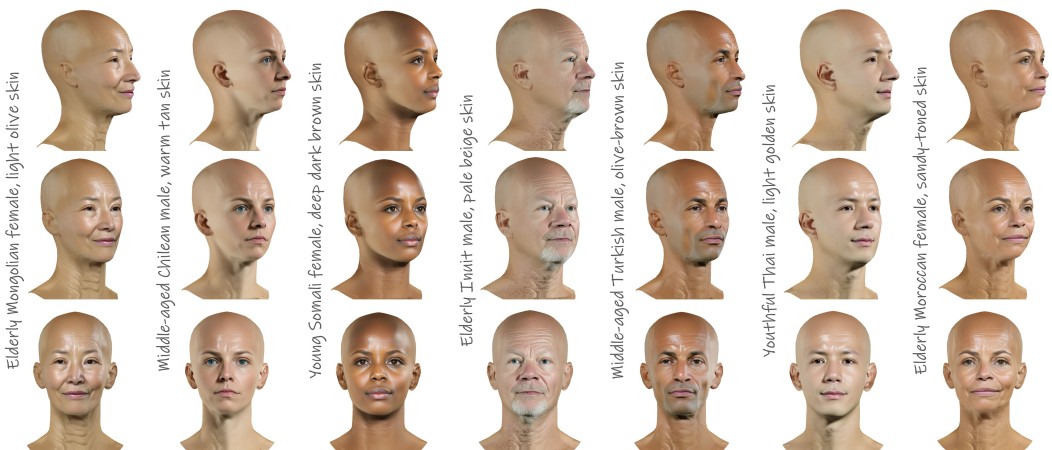

Figure 13: **Photo-Realistic Results (I):** Rendered texture maps of the head are shown from three different views. The textures are displayed across a range of ethnicities and ages, illustrating the versatility and effectiveness of our method in handling diverse facial features with high fidelity. These results correspond to the Figure: "Photorealistic Rendering Results," in the main paper, further demonstrating the robustness of our approach in capturing fine-grained details.

To further evaluate versatility, Figure 19 contains synthesized animal faces, illustrating the method's capability to generate lifelike textures and anatomical consistency. The results reflect detailed fur patterns, expressive facial structures, and species-specific characteristics, demonstrating both realism and artistic stylization.

Finally, Figures 20, 21, 22, 23 and 24 include generations resembling various materials and gemstones, emphasizing the model's ability to synthesize diverse surface qualities. From the translucency of crystals to the roughness of natural stone, the results capture essential visual properties such as light refraction, texture variation, and intricate reflections, reinforcing the adaptability of our technique across different material types.

## A.4 Comparison with 2D Bald Proxy Baselines

In Figures 25 and 26, we compare our approach to several 2D baseline methods, including LDM Inpaint [54], HairMapper [67], and HairCLIPv2 [66]. Our results are shown alongside bald proxy and hairstyle editing methods, demonstrating superior preservation of head shapes and poses while addressing limitations such as quality degradation and inconsistent outputs. This highlights the robustness of our 2D diffusion prior.

## A.5 Texture Maps Results

Finally, Figure 27 provides additional examples of celebrity textures generated from text prompts using fixed meshes. These results include a variety of skin tones, facial features, and expressions, showing the versatility of our method in creating high-quality, photorealistic faces. The generated textures capture the unique facial features that distinguish each celebrity, such as specific bone structures, eye shapes, and other signature traits. By handling a wide range of characteristics, these examples highlight how our approach maintains realism and consistency across different celebrity textures. This demonstrates the robustness of our method in accurately capturing detailed facial features and natural variations.

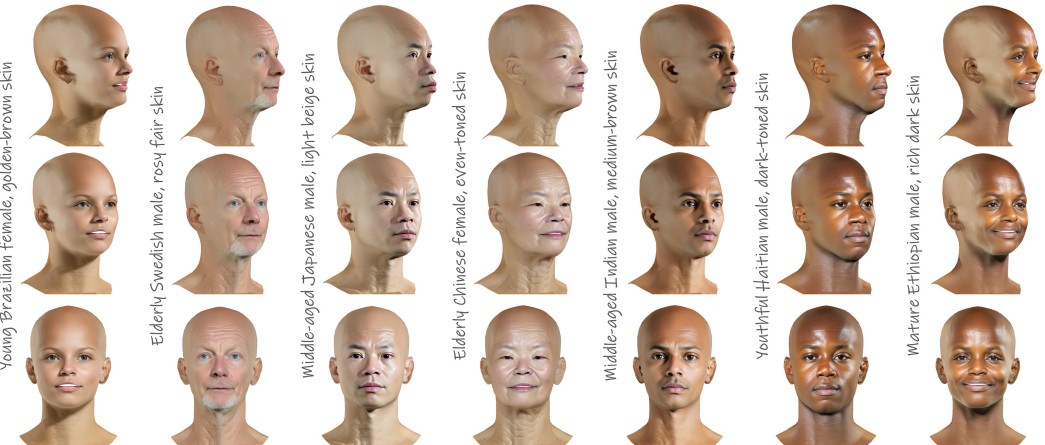

Figure 14: **Photo-Realistic Results (II):** Rendered texture maps of the head are shown from three different views. The textures cover a range of ethnicities, ages, and facial features, demonstrating the versatility and effectiveness of our method in capturing diverse characteristics. These results correspond to the figure "Photorealistic Rendering Results" in the main paper, illustrating the ability of our approach to produce high-quality, realistic faces with consistent detail across views.

| | |
|---|---|
| **A:** *Elderly Mongolian female, light olive skin* | **H:** *Young Brazilian female, golden-brown skin* |
| **B:** *Middle-aged Chilean male, warm tan skin* | **I:** *Elderly Swedish male, rosy fair skin* |
| **C:** *Young Somali female, deep dark brown skin* | **J:** *Middle-aged Japanese male, light beige skin* |
| **D:** *Elderly Inuit male, pale beige skin* | **K:** *Elderly Chinese female, even-toned skin* |
| **E:** *Middle-aged Turkish male, olive-brown skin* | **L:** *Middle-aged Indian male, medium-brown skin* |
| **F:** *Youthful Thai male, light golden skin* | **M:** *Youthful Haitian male, dark-toned skin* |
| **G:** *Elderly Moroccan female, sandy-toned skin* | **N:** *Mature Ethiopian male, rich dark skin* |

Table 4: Skin tones and demographic descriptions for the results shown in the main paper ("Photorealistic Rendering Results").

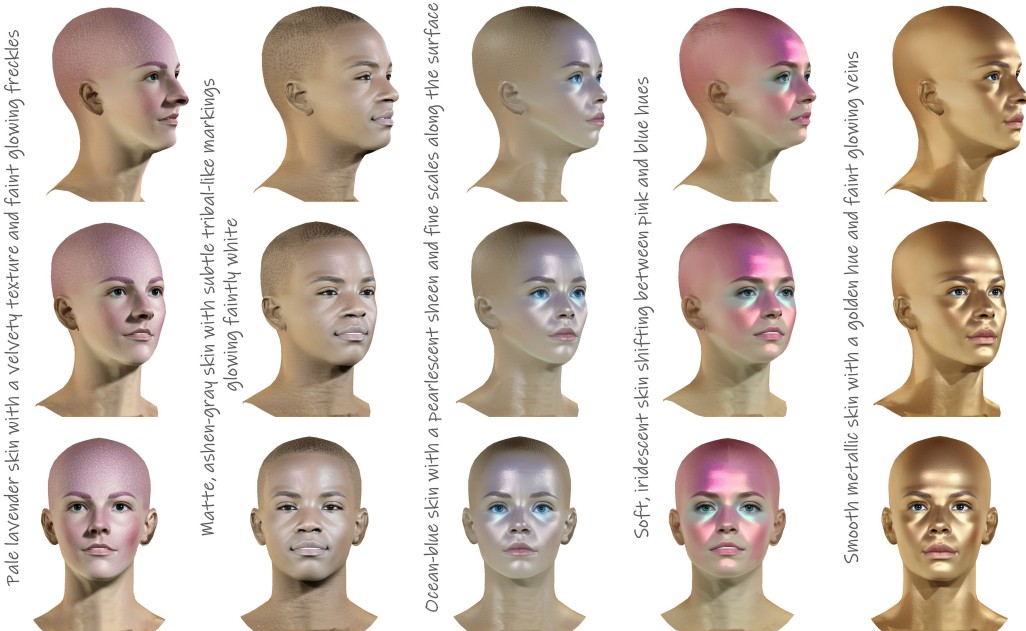

Figure 15: **Abstract (I):** Rendered textures generated from text prompts with "fantasy" elements, shown from three different views. These textures show the consistency and variety of our method in generating imaginative and stylistic facial features.

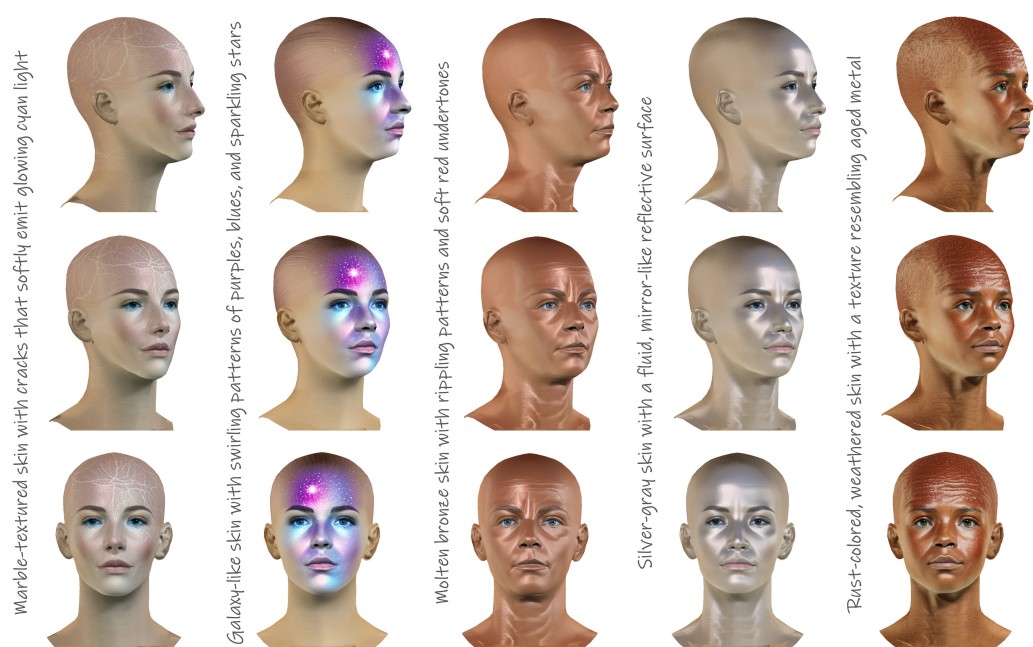

Figure 16: **Abstract (II):** Rendered textures generated from text prompts with "fantasy" elements, shown from three different views. These textures show the consistency and variety of our method in generating imaginative and stylistic facial features.

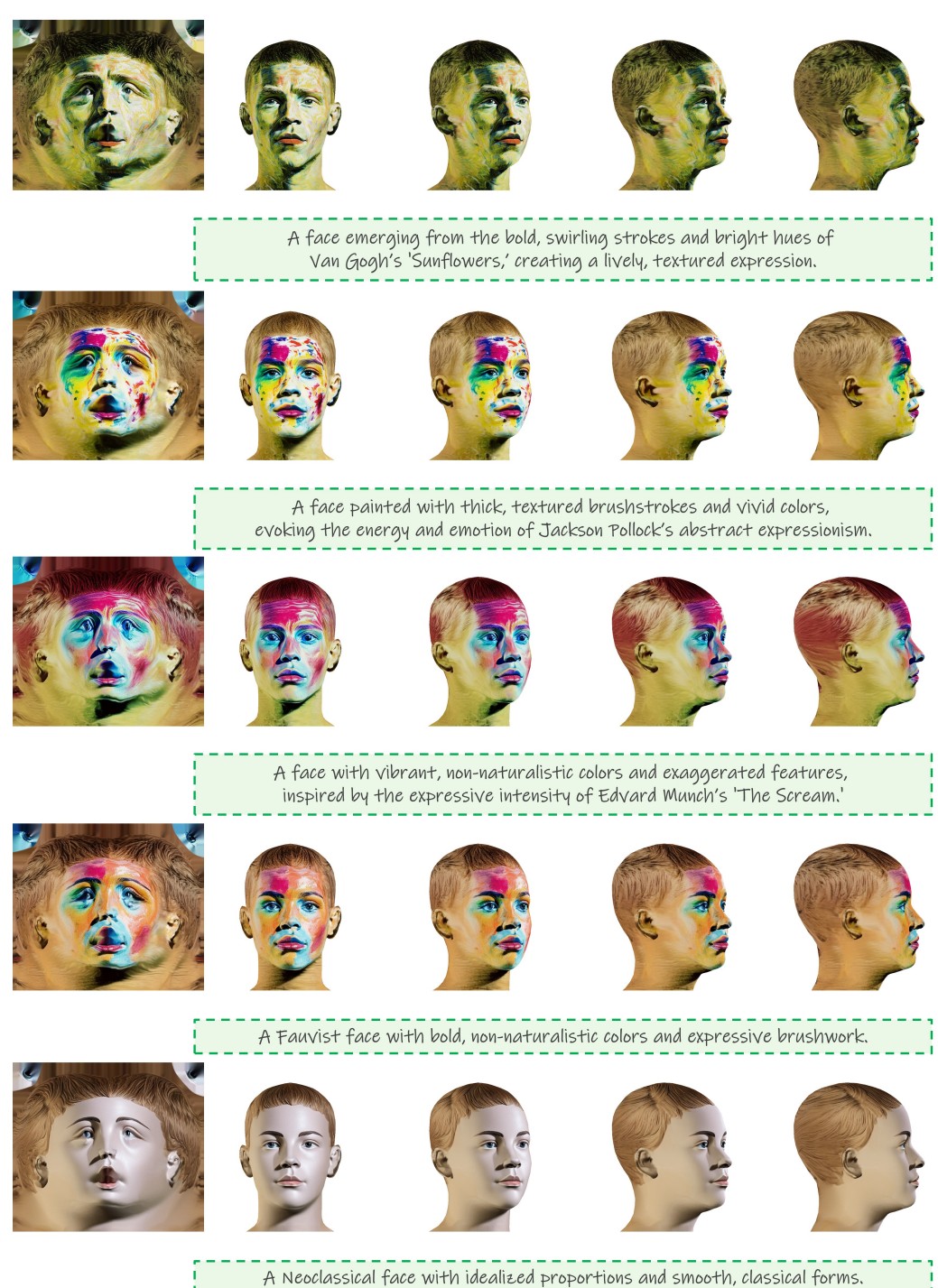

A face emerging from the bold, swirling strokes and bright hues of
Van Gogh's 'Sunflowers,' creating a lively, textured expression.

A face painted with thick, textured brushstrokes and vivid colors,
evoking the energy and emotion of Jackson Pollock's abstract expressionism.

A face with vibrant, non-naturalistic colors and exaggerated features,
inspired by the expressive intensity of Edvard Munch's 'The Scream.'

A Fauvist face with bold, non-naturalistic colors and expressive brushwork.

A Neoclassical face with idealized proportions and smooth, classical forms.

Figure 17: **Painting(s) Styles (I):** Rendered textures generated from text prompts with "fantasy" elements, shown from four different views. These textures show the consistency and variety of our method in generating imaginative and stylistic facial features.

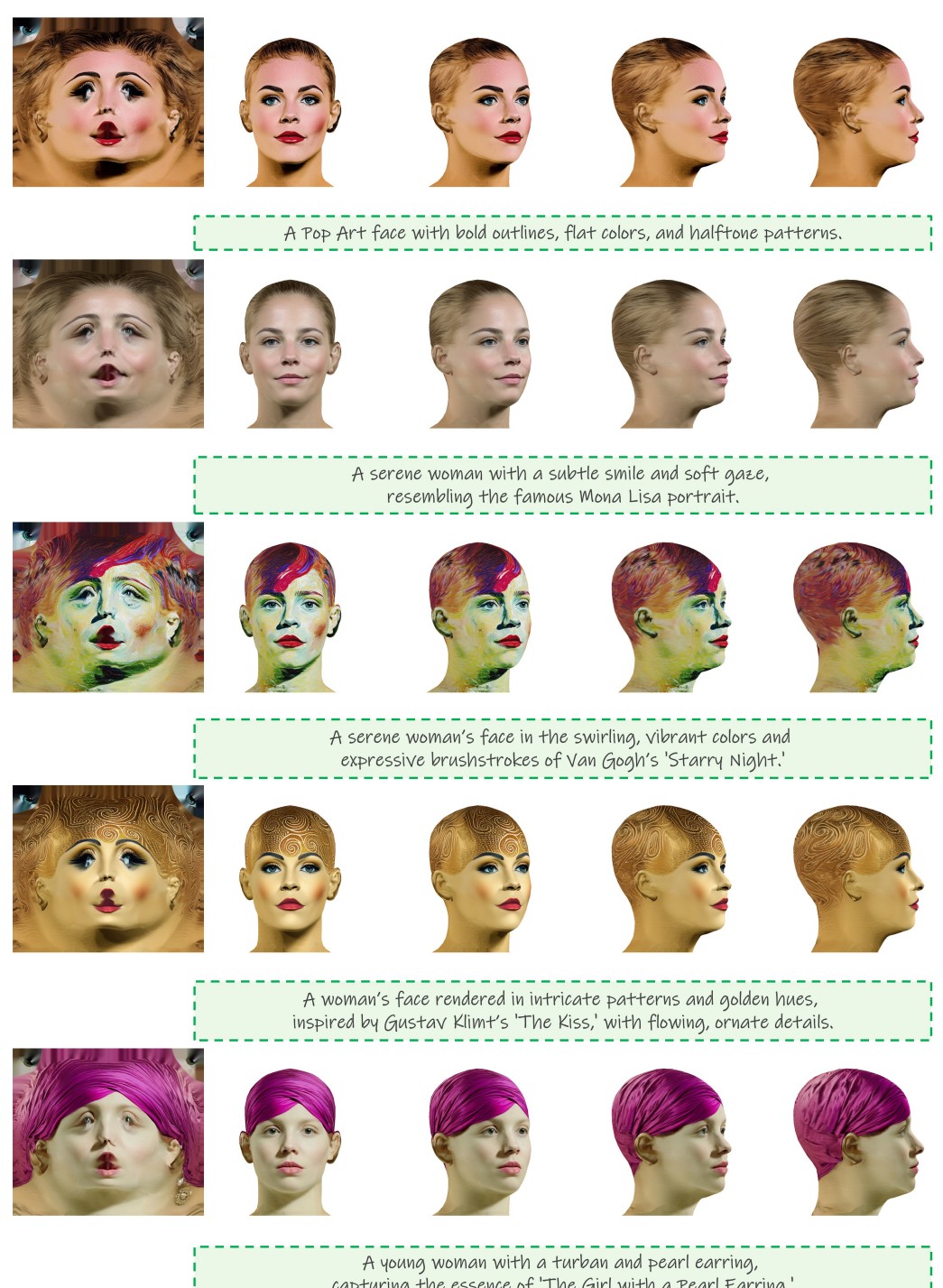

Figure 18: **Painting(s) Styles (II):** Rendered textures generated from text prompts with "fantasy" elements, shown from four different views. These textures show the consistency and variety of our method in generating imaginative and stylistic facial features.

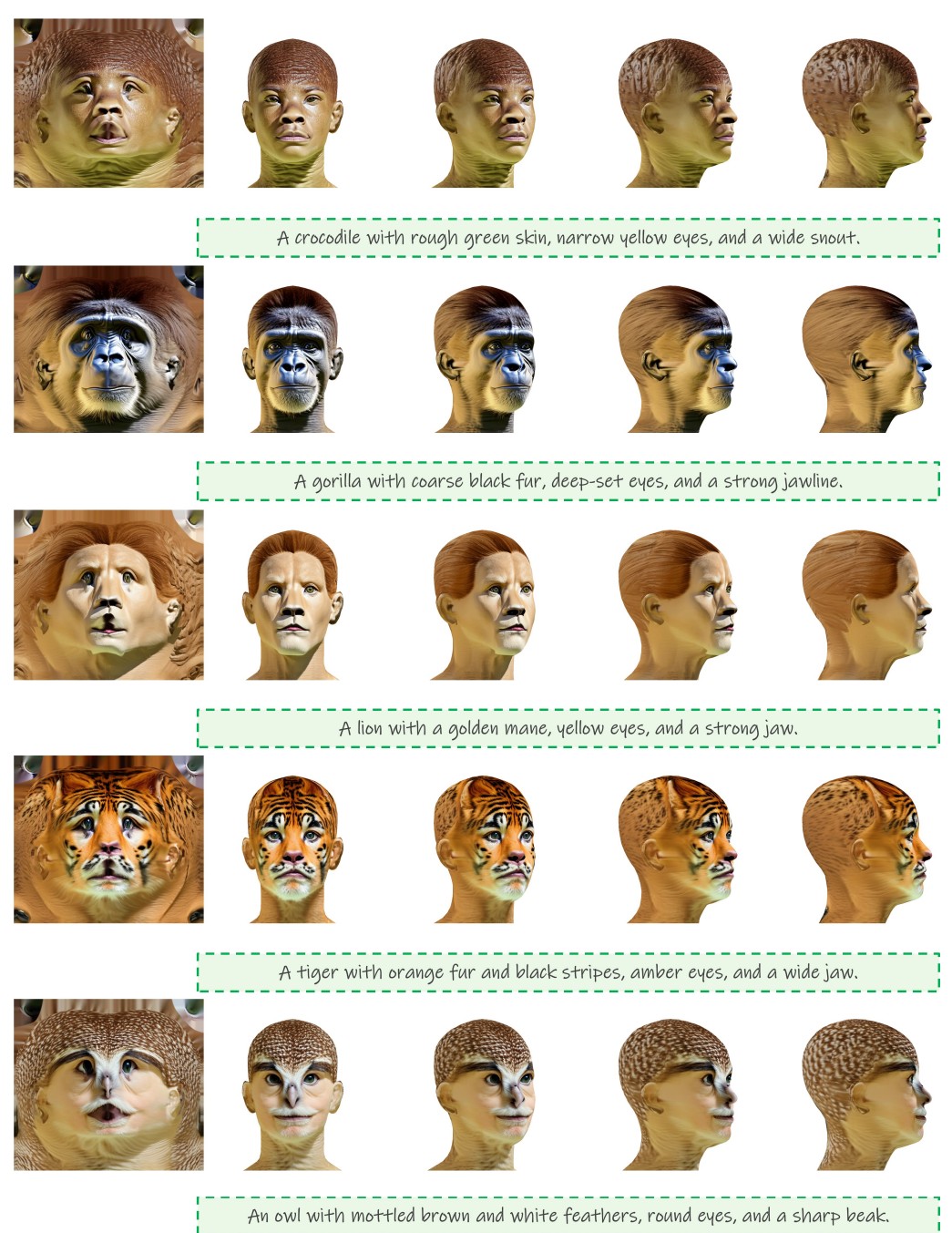

Figure 19: **Animals:** Rendered textures generated from text prompts with "animal" elements, shown from four different views.

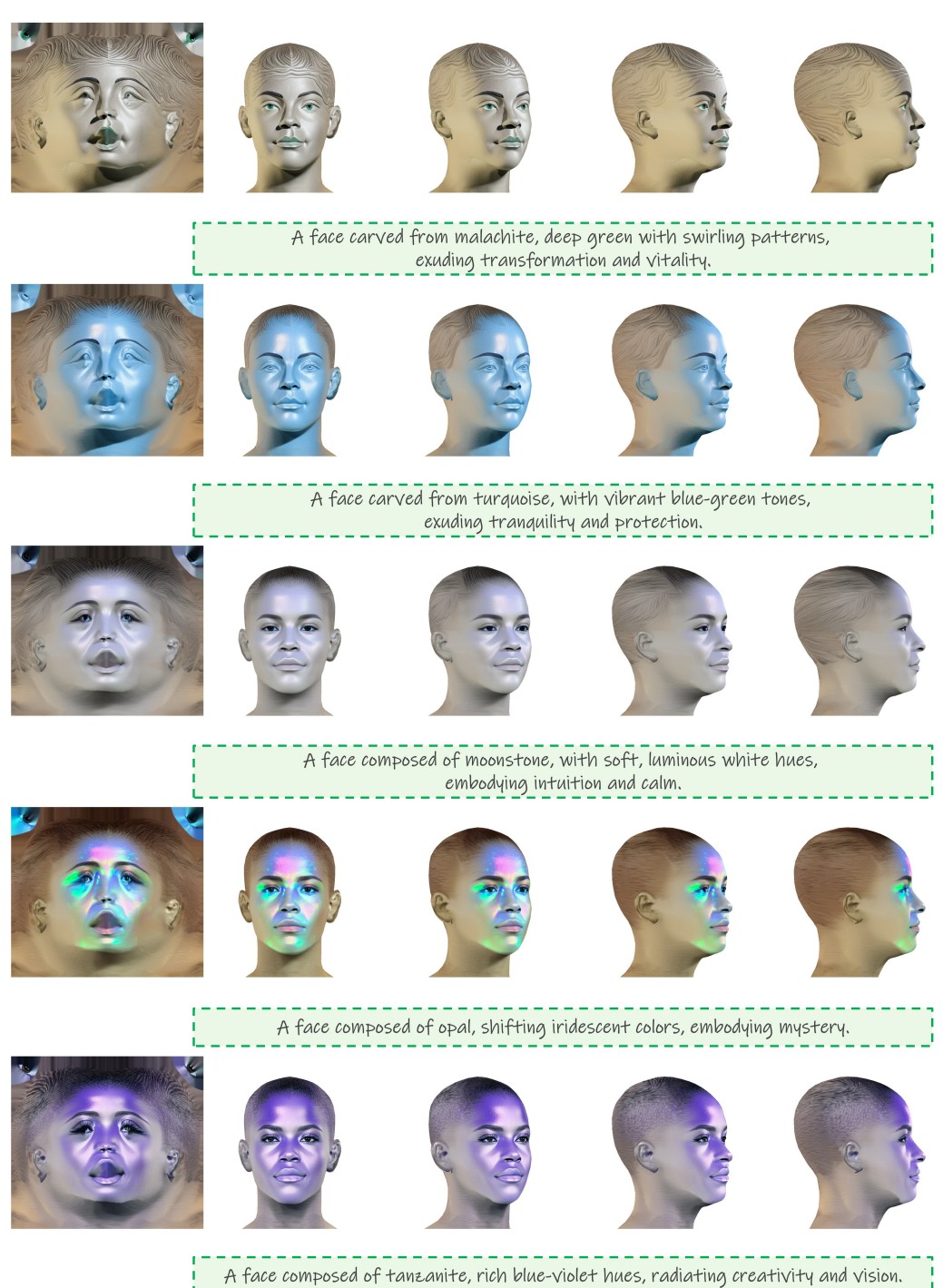

Figure 20: **Gemstones/Materials (I):** Rendered textures generated from text prompts with "gemstone/material" elements, shown from four different views. These textures show the consistency and variety of our method in generating imaginative and stylistic facial features.

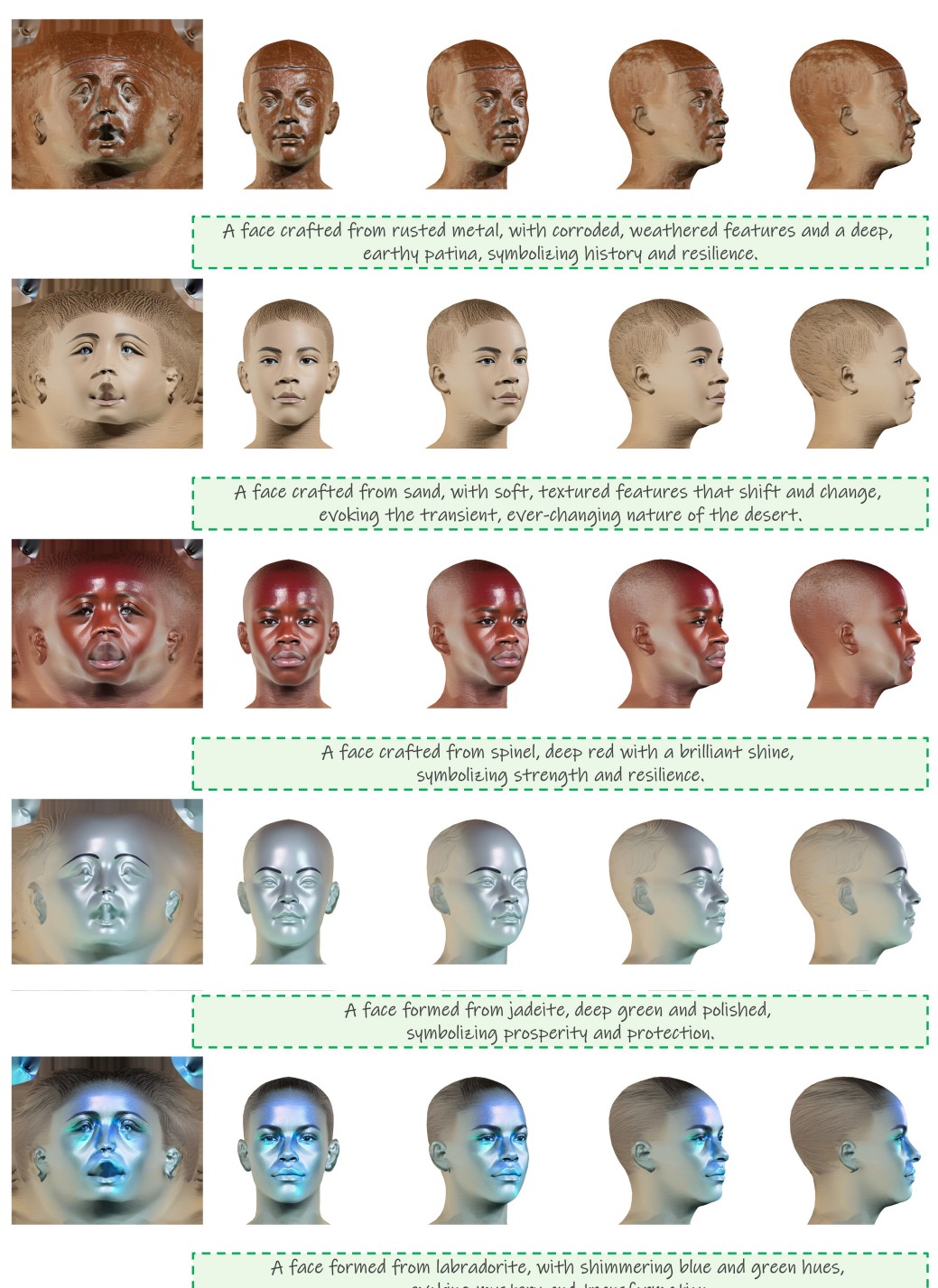

A face crafted from rusted metal, with corroded, weathered features and a deep,
earthy patina, symbolizing history and resilience.

A face crafted from sand, with soft, textured features that shift and change,
evoking the transient, ever-changing nature of the desert.

A face crafted from spinel, deep red with a brilliant shine,
symbolizing strength and resilience.

A face formed from jadeite, deep green and polished,
symbolizing prosperity and protection.

A face formed from labradorite, with shimmering blue and green hues,
evoking mystery and transformation.

Figure 21: **Gemstones/Materials (II):** Rendered textures generated from text prompts with "gemstone/material" elements, shown from four different views. These textures show the consistency and variety of our method in generating imaginative and stylistic facial features.

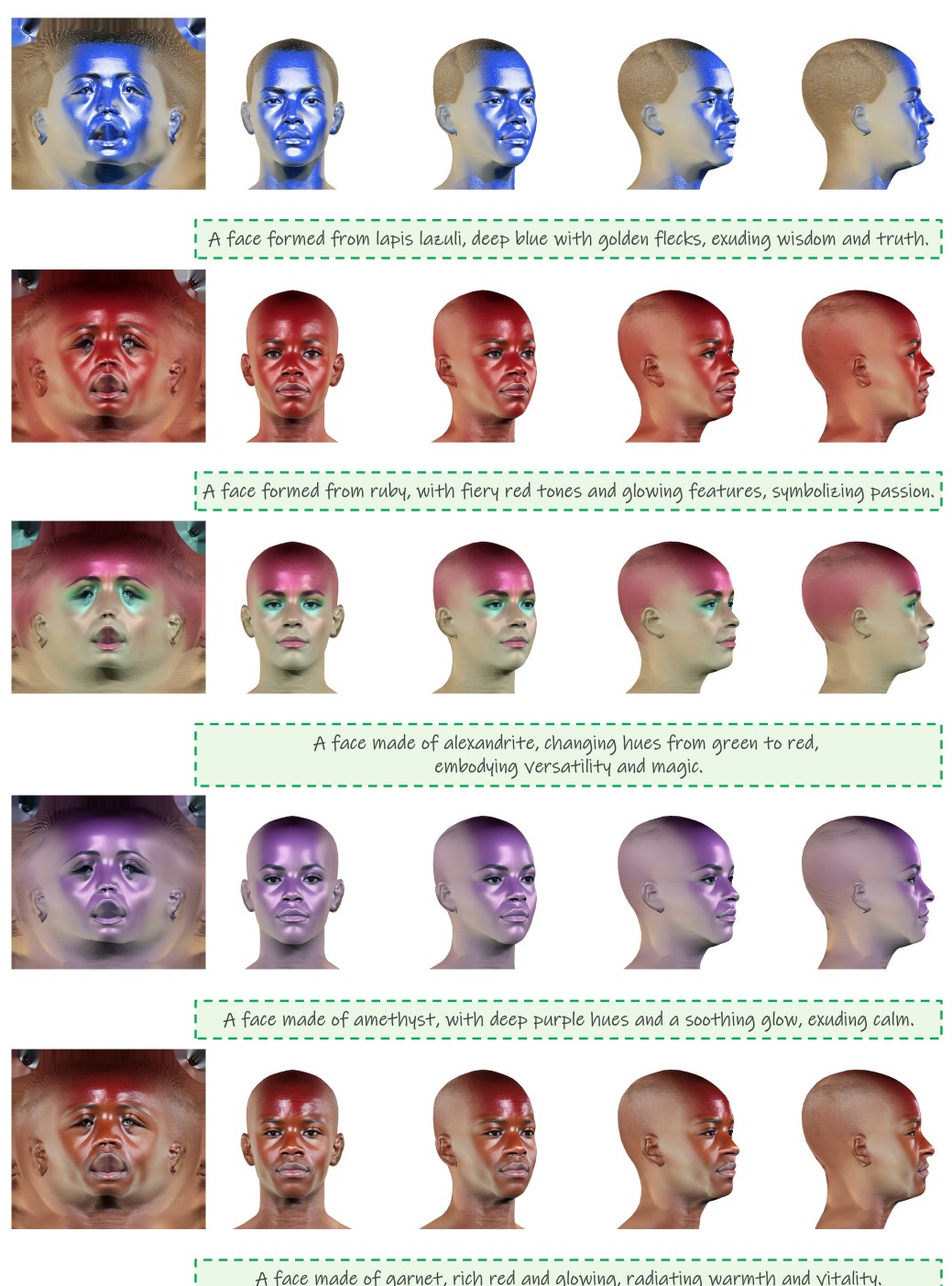

A face formed from lapis lazuli, deep blue with golden flecks, exuding wisdom and truth.

A face formed from ruby, with fiery red tones and glowing features, symbolizing passion.

A face made of alexandrite, changing hues from green to red, embodying versatility and magic.

A face made of amethyst, with deep purple hues and a soothing glow, exuding calm.

A face made of garnet, rich red and glowing, radiating warmth and vitality.

Figure 22: **Gemstones/Materials (III):** Rendered textures generated from text prompts with "gemstone/material" elements, shown from four different views. These textures show the consistency and variety of our method in generating imaginative and stylistic facial features.

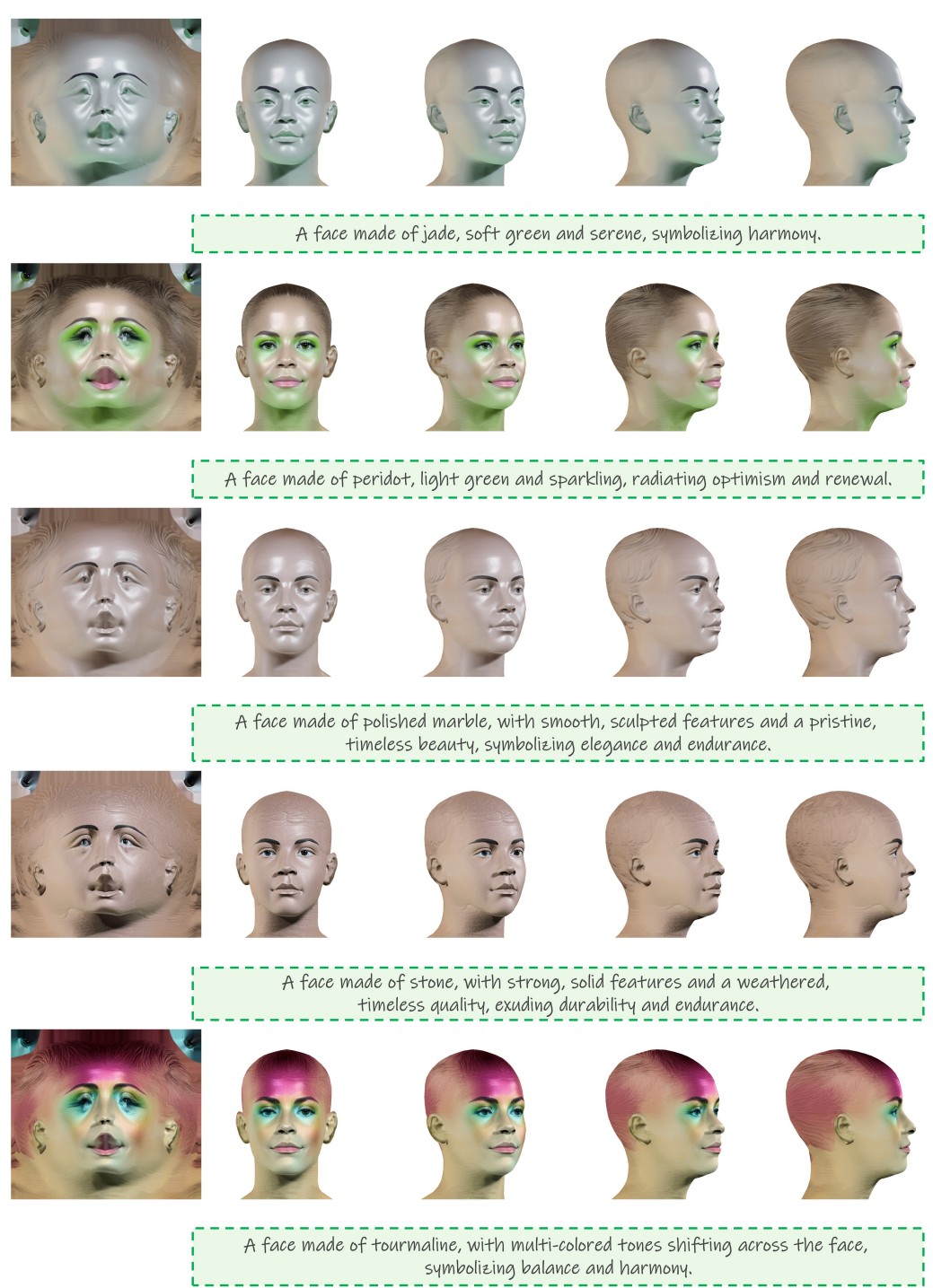

Figure 23: **Gemstones/Materials (IV):** Rendered textures generated from text prompts with "gemstone/material" elements, shown from four different views. These textures show the consistency and variety of our method in generating imaginative and stylistic facial features.

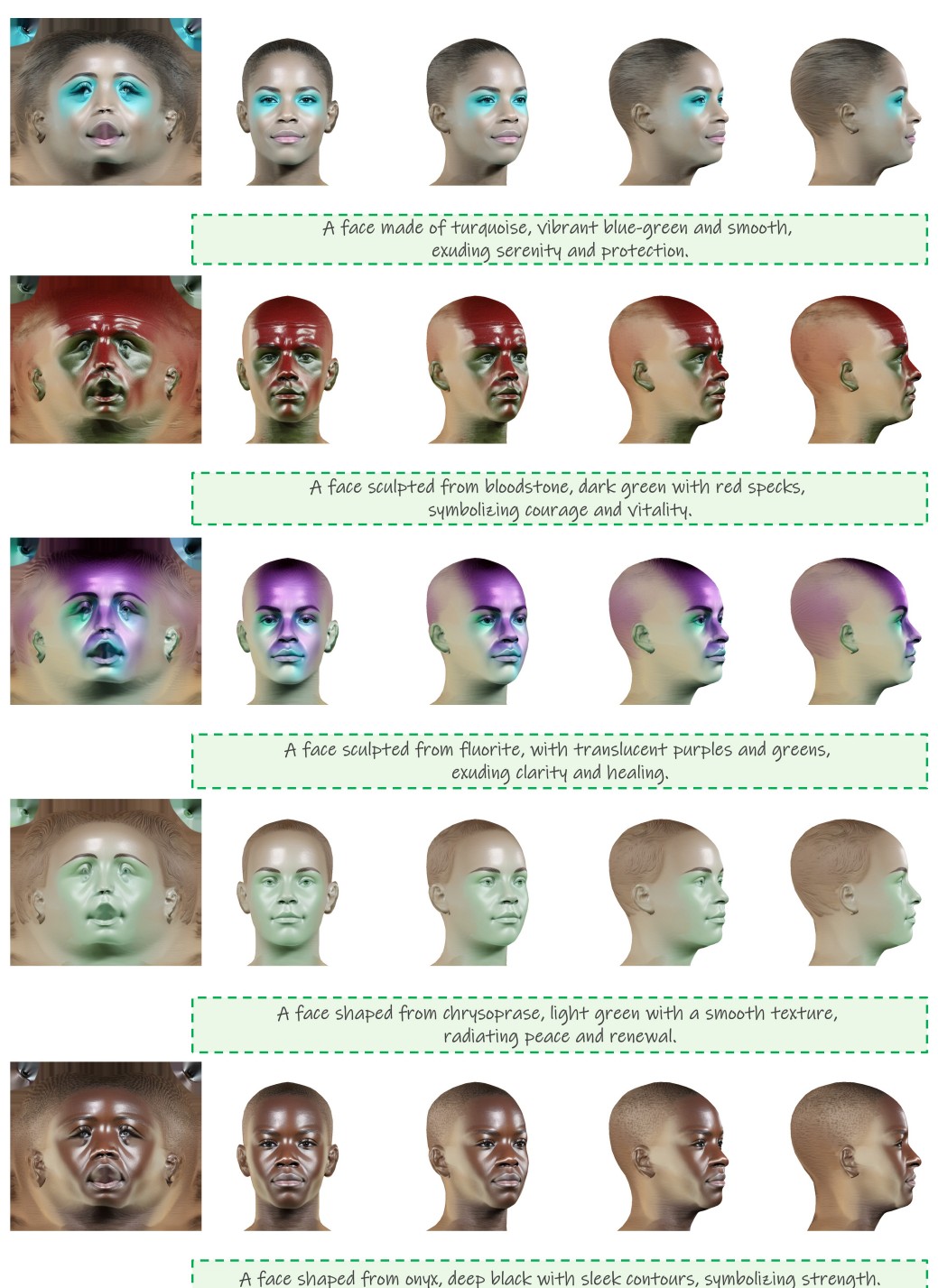

Figure 24: **Gemstones/Materials (V):** Rendered textures generated from text prompts with "gemstone/material" elements, shown from four different views. These textures show the consistency and variety of our method in generating imaginative and stylistic facial features.

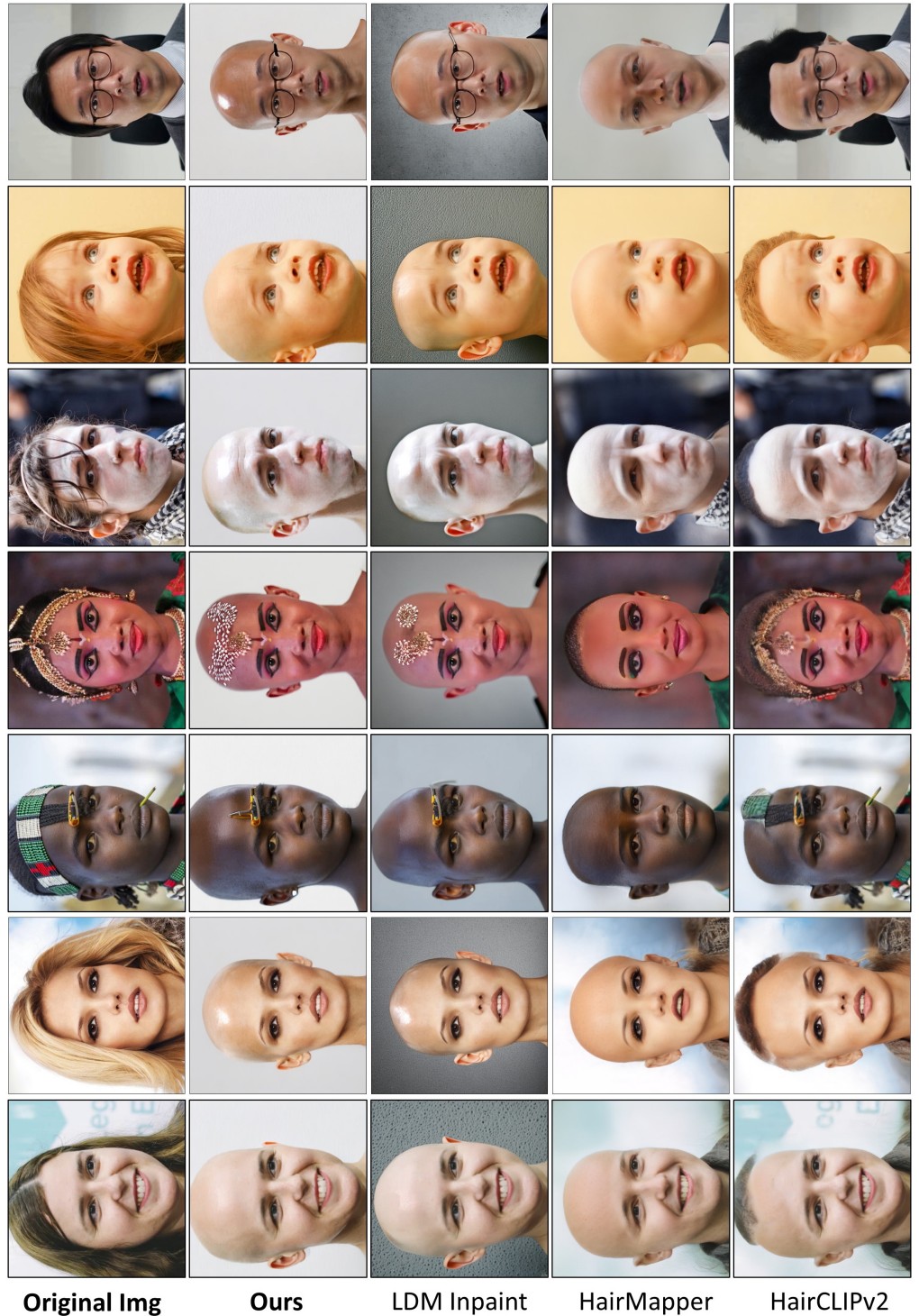

| **Original Img** | **Ours** | LDM Inpaint | HairMapper | HairCLIPv2 |
| --- | --- | --- | --- | --- |

Figure 25: **Comparisons with State-of-the-Art 2D Bald Proxy Methods (I):** Our approach (with CS 0.45) is shown alongside bald proxy and hairstyle editing methods. LDM Inpaint [54] uses the SD 2.1 inpainting model with a full background mask, similar to ours. HairMapper [67] and HairCLIPv2 [66] (prompt: "bald") are bald proxy and hairstyle editing methods, though both degrade image quality; HairCLIPv2 generates hair due to limited bald data in training. Our method addresses this limitation, while preserving head shapes and poses more accurately.

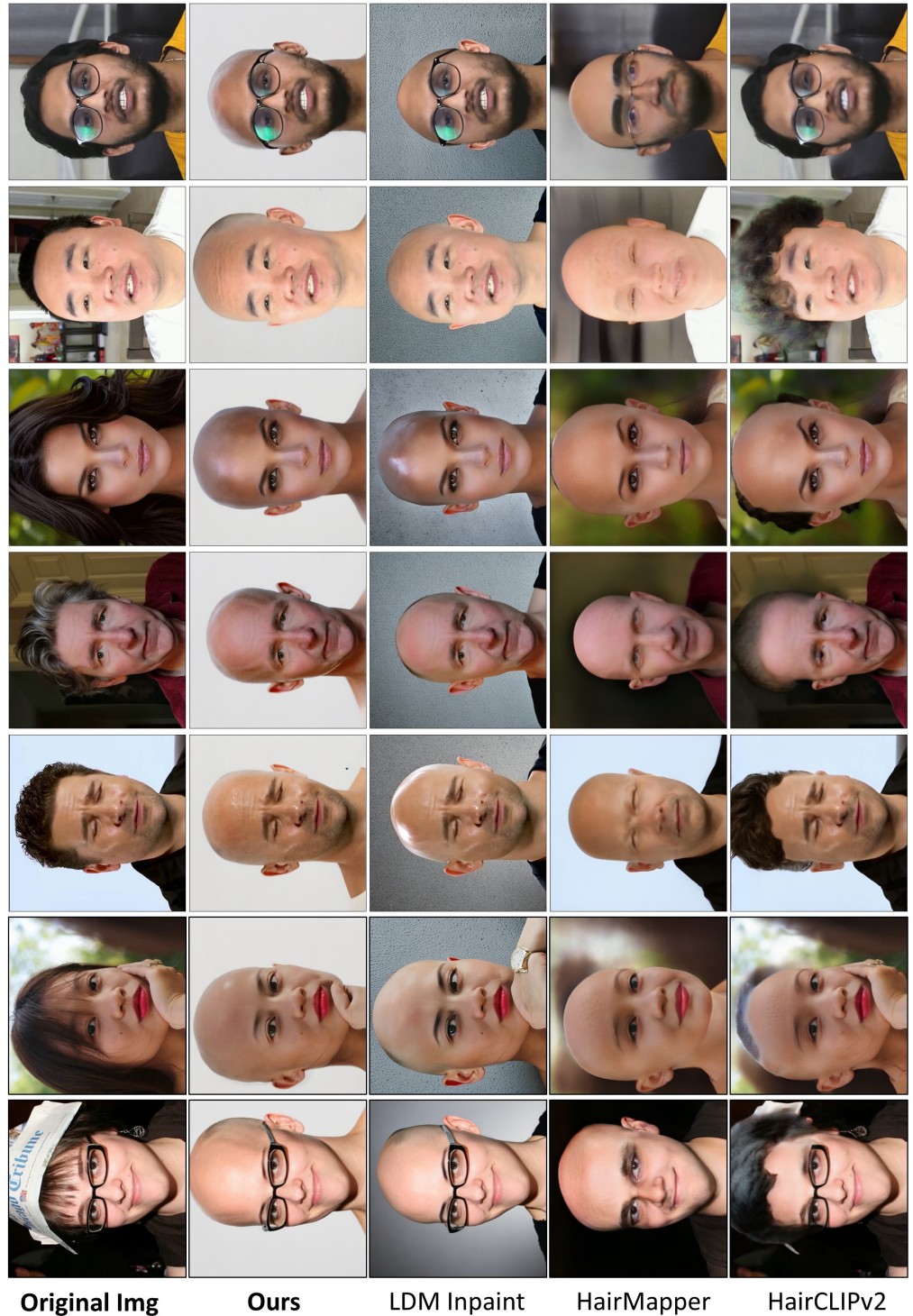

| Original Img | Ours | LDM Inpaint | HairMapper | HairCLIPv2 |

Figure 26: **Comparisons with State-of-the-Art 2D Bald Proxy Methods (II):** Our approach (with CS $0.45$) is shown alongside bald proxy and hairstyle editing methods. LDM Inpaint [54] uses the SD 2.1 inpainting model with a full background mask, similar to ours. HairMapper [67] and HairCLIPv2 [66] (prompt: "bald") are bald proxy and hairstyle editing methods, though both degrade image quality; HairCLIPv2 generates hair due to limited bald data in training. Our method addresses this limitation, while preserving head shapes and poses more accurately.

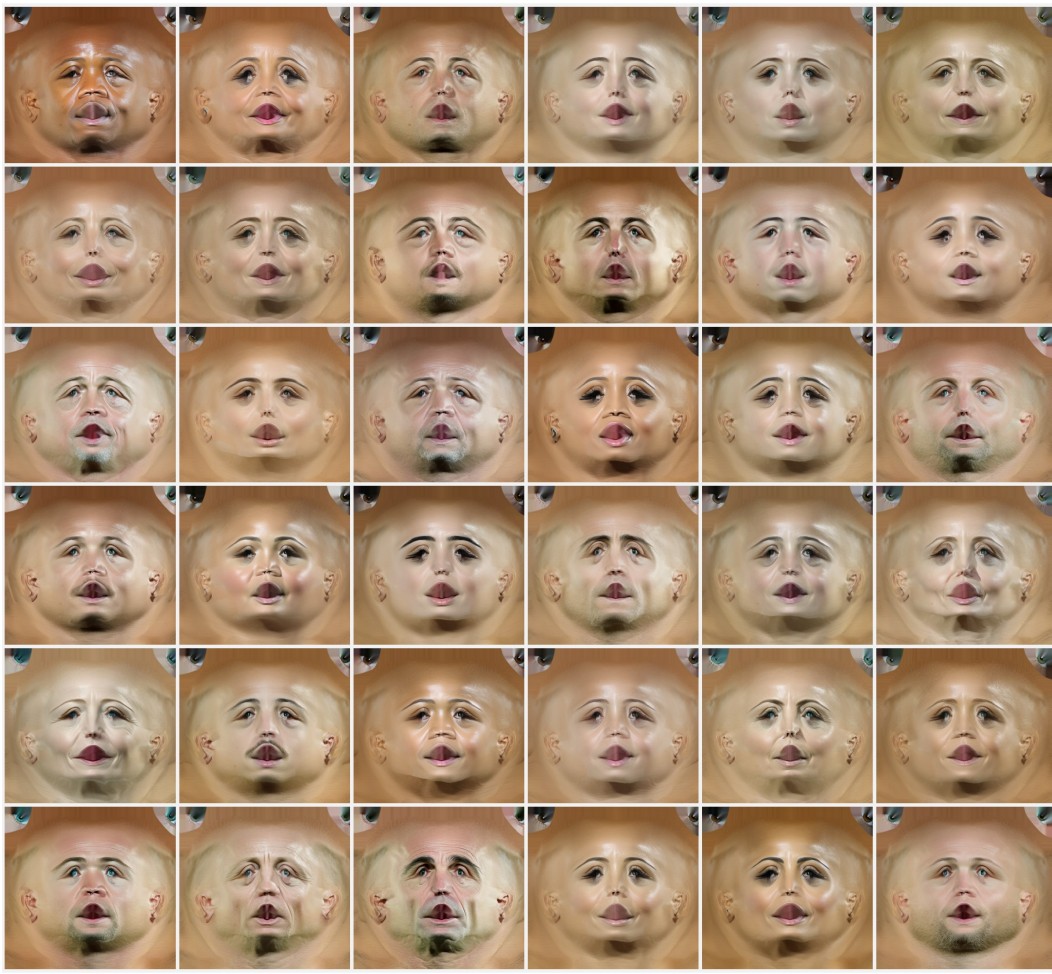

Figure 27: **Additional Texture Maps - Generated Results:** Examples of textures generated from text prompts of random celebrity names. The results include a diverse range of skin tones and facial features, highlighting the effectiveness of our method in generating a large variety of faces.

### A.6    User Study

We ran a perceptual evaluation on Amazon Mechanical Turk to assess avatar plausibility. Each HIT presented a worker with four rendered views (frontal, two sides, and back) of avatars for two different identities, generated by 13 methods. Participants were asked, "Which head avatar looks most plausible?" and were paid $1 USD per completed subject. A screenshot of the study interface is shown in Figure 28.

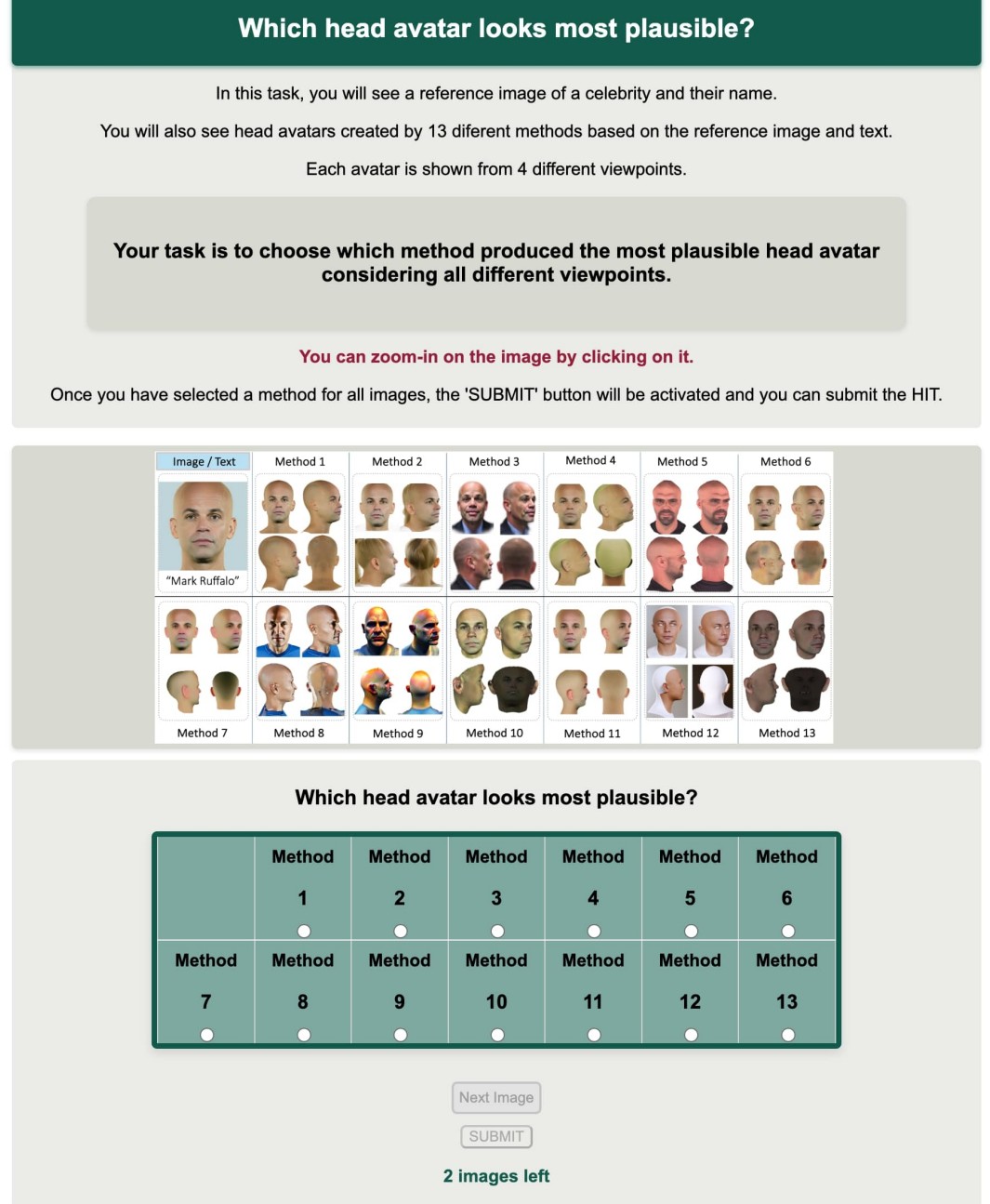

Figure 28: Layout of our user study.

