# OpenReview forum: "HairFree: Compositional 2D Head Prior for Text-Driven 360° Bald Texture Synthesis"
_NeurIPS.cc/2025/Conference — NeurIPS 2025 poster_

### Official Review · Reviewer_ZFQT · 2025-06-03

**Clarity:** 2
**Significance:** 3
**Originality:** 3
**Rating:** 4
**Confidence:** 4

**Summary:**

This paper proposes a novel method for generating bald texture from text conditions. To this end, it trains a compositional 2D head prior that can generate face images given a background image, a set of face parsing masks, and the FLAME mesh as the condition. They further develop a pipeline to create the texture using this 2D prior.

**Questions:**

* About the inpainting network:
  * From the current version, it is very hard for me to figure out the implementation and usage of the inpainting network.
  * In L140, the paper says "we replace the locked LDM with an inpainting variant, allowing for guided completion of missing texture regions" and gives a loss function in Eq.(8). Does this inpainting need training? What is the training data?
  * The word "inpainting network" does not even appear in Sec.4.2. What's the role of the inpainting network in the 3D texturing pipeline?
  * In L207, the paper mentions a baseline called "the model without inpainting". What is it?

* It seems that there is a gap between the training and texturing phase of the 2D prior. In the training phase, there is always a hair mask, but in the texturing phase, the hair mask is set to zero. What about the effects of this gap?

* The training dataset, i.e. FFHQ and CelebA-HQ, does not contain the top view of bald people. How does the method synthesize the top view of the bald head? In addition, how does the method synthesize the texture of the back head?

As there is so much missing information in the current version, I lean towards rejecting this paper.

**Ethical Concerns:**

["NO or VERY MINOR ethics concerns only"]

**Final Justification:**

All my concerns are addressed in the rebuttal. I'm now leaning positively toward this paper. I do not give a higher score as I think the technical contribution is somewhat limited.

**Limitations:**

yes

**Quality:**

2

**Strengths And Weaknesses:**

### Strengths:
* The comparison experiments are solid. Many relevant works are involved for comparison. The proposed method indeed obtains the best results.
* The proposed compositional 2D head prior will serve as a foundation model in the field of face generation, if the model weights can be released.

### Weaknesses
* The paper writing does not meet the bar for publication. See the questions in the section below.
* The dataset is one of the main contributions listed in the introduction. However, there are no descriptions for this dataset in the main paper! Thus, it is hard for me to evaluate the significance of this dataset. There are also many datasets in the community, including the FFHQ-UV and the FFHQ-UV-Intrinsics dataset. If the paper claims its dataset is a contribution, it should be compared to existing datasets explicitly and clarify what the benefit of their dataset is over the existing ones.

---

> ### Author Rebuttal · Authors · 2025-07-31
>
> Thank you for providing constructive feedback on how to improve the presentation and writing of the paper.
>
>
> We are grateful to hear that the compositional 2D head prior will be useful for the community (yes, we will release the model weights, see line 19) and that the experiments are solid, showing that the proposed method obtains the best results.
>
>
> In the following, we will provide the additional requested details which we will also add to the revised paper.
>
>
> **(1) Implementation / Usage of the inpainting network:**
>
>
> - Does this inpainting need training? What is the training data?
>
>   - The inpainting LDM prior is functionally equivalent to the locked (frozen) LDM prior, with the key distinction that the inpainting model was additionally conditioned on masks during training. In our setup, we do not retrain the inpainting LDM; rather, we use a pre-trained version and replace the original LDM prior with it. At inference time, we attach ControlNet to the inpainting LDM instead of the original LDM, allowing us to perform inpainting using both mask-based conditioning and ControlNet guidance. For further details on the training of prior LDMs, we refer the reader to Rombach et al. [51] (Stable Diffusion).
>
>
> - What's the role of the inpainting network in the 3D texturing pipeline?
>
>   - At inference time, we use the inpainting LDM prior in combination with our ControlNet to generate head images conditioned on view-dependent 3D head mesh and skin/ear appearance cues. As the camera gradually rotates around the head, previously unseen head regions become visible and are inpainted by the model. The visible regions that have already been synthesized are stored in UV space and later re-rendered and reused for inpainting as the camera continues to move, ensuring spatial and temporal consistency across views. (See Figure 2 (system figure) and its description, and Sec. 4.2).
>
>
> - What is the baseline called "the model without inpainting"?
>
>
>   - The "model without inpainting" generates all head views at the same time, without accumulating previously seen regions in UV space or inpainting newly visible areas. In contrast, our progressive texturing approach is iterative: views are generated one after another and depend on previously generated content. As the camera rotates, visible regions are stored in UV space, and newly revealed areas are inpainted based on both the current view and prior results. This approach better handles extreme angles outside the fine-tuning distribution.
>
>
> **(2) It seems that there is a gap between the training and texturing phase of the 2D prior. In the training phase, there is always a hair mask, but in the texturing phase, the hair mask is set to zero. What about the effects of this gap?**
>
>
> Yes, the mask distribution in the training and texturing phase is different. During training, ControlNet learns hair appearance conditioned on the hair mask. At test time, by setting the hair mask to zero, reducing the ControlNet strength to rely more on the inpainting prior, and specifying via text that the head should be bald, the model efficiently removes hair. Combined with our progressive texturing approach, this enables plausible bald head generation mitigating artifacts coming from the training/inference conditioning gap.
>
>
> **(3) The training dataset, i.e. FFHQ and CelebA-HQ, does not contain the top view of bald people. How does the method synthesize the top view of the bald head? In addition, how does the method synthesize the texture of the back head?**
>
>
> Since the training data lacks top and back views of bald heads, we reduce the ControlNet strength to enable the generation of these views. This allows the model to rely more on the inpainting LDM prior, which was trained on larger, more generic datasets, making the fine-tuned dataset less critical. Combined with our progressive texturing approach and a text prompt specifying a bald head, this helps produce plausible textures for extreme angles (See lines 237-238 and Figure 6(C) for Controlnet strength; lines 140-143 for the inpainting LDM; lines 168-189 and Figures 6(A) and Figure 8 for progressive texturing and related ablations).
>
>
> **(4) Dataset contribution:**
>
>
> We would like to clarify that the dataset itself is not the primary contribution of our work. Rather, our key contribution lies in the methodology that enables the generation of high-quality, 360-degree bald head textures. This method can serve as a powerful tool for data generation in downstream tasks. Photorealistic examples from the resulting dataset are presented throughout the paper and in the Appendix. Importantly, our dataset is generated entirely automatically using our method and can be scaled or reproduced using only text prompts and 3D head meshes. This makes it fundamentally different from datasets such as FFHQ-UV, which rely on GAN-based models, produce lower-quality textures, and include baked-in hair in the generated outputs. Additionally, FFHQ-UV requires a reference image and does not allow control over the underlying head geometry.
>
>
> Thank you again for your time and constructive feedback. We hope our responses have clarified the key points, and we would be glad to provide further details if needed.

---

> > ### Comment · Reviewer_ZFQT · 2025-08-04
> >
> > Thanks for the rebuttal. My main concerns have been addressed. I will discuss to AC and other reviewers and give a final score.

---

> > > ### Author Response · Authors · 2025-08-05
> > >
> > > Thank you for your response. We're pleased to hear that your main questions have been addressed. Should any additional points arise, please don’t hesitate to reach out.

---

### Official Review · Reviewer_gR5R · 2025-06-17

**Clarity:** 3
**Significance:** 2
**Originality:** 2
**Rating:** 4
**Confidence:** 4

**Summary:**

This paper proposes a method to synthesize RGB head textures from text prompts. The proposed method first trains a ControlNet, conditioned on masks and a rendered face mesh. It then generates multiview images from a text prompt. It starts from the frontal image and sequentially infills the remaining texels from the other views. The paper conducts a visual comparison with a dozen related works and a user study on two samples. An ablation study measures the effect of the guidance scale.

**Questions:**

- What do Person 1 and Person 2 refer to in the user study? How were these examples chosen? Are these two samples representative of the output distribution?
- How would the proposed texturing aggregation pipeline compare to classical texture fusion approaches (Ebert et al. 2002) or more recent methods like Haoran et al. (2023) and Zheng et al. (2023)?
- Why does the proposed texture aggregation method handle seams better than related works? It would be valuable to provide more and higher resolution examples of the back of the head, where FLAME has a major seam. There are a few examples in Fig. 3 but they are very small. Providing higher resolution images and a discussion why this approach improves over existing methods would be appreciated.


References: See weaknesses

**Ethical Concerns:**

["NO or VERY MINOR ethics concerns only"]

**Final Justification:**

The contributions are not very innovative, but the contributions are technically solid and carefully evaluated.

**Limitations:**

yes

**Quality:**

2

**Strengths And Weaknesses:**

## Strengths

- The paper compares with 12 recently published works.
- The visual results look promising, in particular for the back of the head where FLAME has a major seam.
- Quantitatively evaluating results for the given problem (texture generation from text prompts) is difficult, and I appreciate that the authors made an effort to conduct a user study.

## Weaknesses
- Limited ablation study.
     - The only ablation provided measures the influence of the guidance scale. It concludes that the guidance scale trades visual fidelity with diversity---a fact already established in previous works (Ho and Salimans 2022). I would have liked to see an ablation study that explains why the proposed texture aggregation method handles seams better than existing techniques like (weighted) averaging or interpolations (Ebert et al. 2002).
- Minor contributions
    - The paper makes two contributions: a ControlNet-based 2D head prior and an iterative texturing approach. The first contribution is a straightforward application of ControlNet. The second contribution is an interactive texture-infilling, which is very closely related to existing literature (Ebert et al. 2002, Haoran et al 2023, Zheng et al. 2023). It would be great if the paper could describe the differences to the above texturing unprojection techniques and, ideally, include a comparison.
- References
    - Ho, Jonathan, and Tim Salimans. "Classifier-free diffusion guidance." __arXiv preprint arXiv:2207.12598__ (2022).
    - Ebert, David S., et al. __Texturing and modeling: a procedural approach__. Elsevier, 2002.
    - Bai, Haoran, et al. "Ffhq-uv: Normalized facial uv-texture dataset for 3d face reconstruction." __Proceedings of the IEEE/CVF conference on computer vision and pattern recognition__. 2023.
    - Zheng, Wenzhuo, et al. "Multi-view 3d face reconstruction based on flame." __CoRR__ (2023).

---

> ### Author Rebuttal · Authors · 2025-07-31
>
> Thank you for your valuable feedback and highlighting the strengths of our work!
>
>
> In the following, we give additional insights and clarify questions raised:
>
>
> **(1) Our contributions:**
>
>
> As mentioned, our approach is able to generate diverse textures without any major seams. This is achieved through our proposed compositional 2D prior and a progressive texture merging module that ensures consistency across views. While both components build on prior ideas, they are combined and adapted in a novel way to enable fully automatic, 3D-consistent 360° head texturing for both human and non-human skin—without requiring reference images, multi-view data, or bald human head data.
>
> - The ControlNet-based 2D head prior is trained to generate semantically aligned RGB views, conditioned on 3D geometry and face parsing. As noted by Reviewer ZFQT, this component acts as a strong foundation model for face generation, supporting a wide range of skin types and enabling robust hair-skin separation for bald texture synthesis.
>
> - The progressive texture merging module incrementally completes the UV texture by re-rendering and infilling texels from new viewpoints, using previously generated texels as a reference.
>
> These design choices are key to the robustness and generality of our method across identities, viewpoints, and texture styles. Moreover, quantitative and qualitative evaluations show that our method outperforms all recent state-of-the-art techniques in terms of texture quality, consistency, and realism.
>
>
>
> **(2) What do Person 1 and Person 2 refer to in the user study? How were these examples chosen? Are these two samples representative of the output distribution?**
>
>
> Person 1 and Person 2 refer to the individuals shown in Figure 3. We will clarify this in the paper. These examples represent photorealistic results and were selected as a random male and female celebrity. For additional examples of this type, please refer to Figure 4. For a broader set of celebrity texture maps, see Appendix Figure 23.
>
>
> **(3) Why does the proposed texture aggregation handle seams better than existing techniques?**
>
>
> - The related methods mentioned are based on fixed inputs, i.e., multiple captured images that might have variations in shading or even facial expressions. In contrast, HairFree is generating its views with a technique that ensures consistency in image space which results in consistent and seam-less uv textures. Specifically, our proposed inpainting technique ensures to use the already existing colors from the texture and generates new color values for regions that are becoming visible when rotating the 3D head in image space. In the paper, we are visualizing the views that are iteratively generated with a major order of front, sides, back. This order also contributes to the consistent merging of the left and the right sides.
>
>
> - Note that we have conducted qualitative ablation studies on inpainting in UV space (inpainting in UV space leads to the seams/discontinuities at the back/top of the head, see Figure 8) and inpainting in RGB space (without our progressive re-rendering approach, it is not possible to generate 3D head-consistent extreme side views and back view; notably, Janus effect will appear at the back, see Figure 6(A)).
>
>
> - We will add additional zoom-ins to the figures that show the head from the back. Note that larger examples, additional texture maps, and more extreme viewpoints are provided in the Appendix. We recommend zooming in on the back view samples for a clearer inspection.
>
>
> **(4) Influence of guidance scale:**
>
>
> Ho and Salimans do not measure the influence of guidance scale when the large diffusion prior is fine-tuned on a smaller dataset. In our case, we fine-tune our prior on photorealistic frontal views of faces only, and at inference time, we show that we can generate back-views, bald samples, and non-photorealistic examples that are out of the fine-tuning distribution, emphasizing strong generalization properties of our approach.
>
> Thank you again for your time and constructive feedback. We hope our responses have clarified the key points, and we would be glad to provide further details if needed.

---

> > ### Comment · Reviewer_gR5R · 2025-08-05
> >
> > Thank you for the clarifications and the promise to add additional zoom-ins for the back of the head.
> >
> > Why does the Janus effect when running without progressive inpainting? The ControlNet is conditioned on semantic maps from face parsing (Eq. 5). Shouldn't these semantic inputs avoid the Janus effect?
> >
> > Similar to 9xQ3, I am still concerned about the novelty of the proposed method. The paper and rebuttal do not discuss the positioning with respect to classical texture fusion approaches (Ebert et al. 2002) and more recent methods like Haoran et al. (2023) and Zheng et al. (2023).

---

> > > ### Author Response · Authors · 2025-08-06
> > >
> > > Thank you for reading the rebuttal texts and reaching out for additional clarifications. Please find our response below:
> > >
> > > **(1) Why does the Janus effect occur without progressive inpainting, even with semantic conditioning?**
> > >
> > > The Janus effect arises because the ControlNet is trained mostly on frontal and side views. When directly conditioned on a back-view mesh and semantic mask, especially with the default ControlNet strength of 1.0, it tends to hallucinate a face, having never seen such inputs during training.
> > >
> > > Progressive inpainting resolves this by gradually completing the texture from front to back, so later views only need to fill in small missing regions. Lowering ControlNet strength during these steps prevents over-conditioning on unfamiliar views.
> > >
> > > Additionally, we use a single face and skin mask that combines all facial attributes (eyes, lips, nose, etc.) into one region. We avoid using separate masks for each part, as many of these features are small and prone to inaccuracies in the off-the-shelf face parsing model. See “Devised Input Conditioning” in Figure 2 for an example of this mask.
> > >
> > > **(2) Discussion and comparison of related work:**
> > >
> > > As described in (3), the classical texture fusion approaches require consistent input views and are not designed for progressive content generation. In contrast, our texturing procedure is designed to work with our image-space inpainting module. Together with our learned image prior, these are the key contributions that enable the generation of bald textures suitable for compositional avatars.
> > >
> > > - *Ebert et al. 2002*:  In the book ‘Texturing & Modeling’, procedural texturing methods are described. Procedural texturing is primarily used to generate material textures such as ‘marble, wood, and stone’ from noise, as well as content like clouds, as the authors note. These textures typically consist of regular patterns and are not designed to generate more complex structures like an entire face. However, procedural modeling could be applied at a micro level, for example, to generate fine details such as skin pores.
> > >
> > > - *Haoran et al. 2023*: In ‘Ffhq-uv: Normalized facial uv-texture dataset for 3d face reconstruction’, partial skin observations from the front are merged with a template texture using pyramid blending. Pyramid blending is a classical method that compensates for input inconsistencies, such as color shifts, by operating in gradient space to transfer detail from partial observations to the template texture. In our approach, consistency is ensured during the generation process itself (via inpainting of novel views); therefore, we do not require a gradient-based merging technique. Additionally, we do not rely on a predefined template texture. Instead, we progressively generate all necessary views to create a complete 360° texture. Note that the template texture used in Haoran et al. also limits the variation of actual textures that are generated (e.g. all textures have the same back of the head / hairstyle).
> > >
> > > - *Zheng et al. 2023*: In ‘FLAME-based Multi-View 3D Face Reconstruction’, a face reconstruction pipeline is described. This is different from our texture generation pipeline, where the 3D mesh is given and views are generated to be consistent. It is not described which fusion mechanism is used to project the input image to their reconstructed FLAME-mesh (‘We extract the texture of multi views and perform simple fusion to obtain I′uv, which contains information from multi views.’, page 5 of the paper).
> > >
> > > We welcome any further questions, particularly those related to the discussion of related work. Thank you!

---

> > > > ### Comment · Reviewer_gR5R · 2025-08-08
> > > >
> > > > Thank you for clarifying the positioning and why the Janus effect happens.

---

### Official Review · Reviewer_9xQ3 · 2025-06-20

**Clarity:** 2
**Significance:** 2
**Originality:** 2
**Rating:** 4
**Confidence:** 3

**Summary:**

This paper proposes a compositional 2D head prior for text-driven 360^{o} bald texture synthesis. This method mainly includ etwo modules: finetuning controlnet with face condition and generate head texture with progressive uv mapping. Comparison with competitors in this work have shown the superiority of proposed pipeline.

**Questions:**

Please see weakness, A table or figure showing performance differences when each conditioning signal is removed would help clarify whether each is necessary. This would strengthen claims of methodological contribution. Also,  Could the authors provide implementation details such as architecture configs, training time, and compute resources needed for reproduction?

**Ethical Concerns:**

["NO or VERY MINOR ethics concerns only"]

**Final Justification:**

Thanks for the response. Most of my concerns have been addressed. My remaining concern is the necessity of incorporating various signals. I believe there is redundant information among the skin and face mask, as well as the rendered 3D head mesh. While I appreciate the systematic design for tackling the human head texturing task, the composition of existing techniques still lacks strong methodological novelty. The authors are encouraged to release the code to make the design choices in each part clearly.

**Limitations:**

Yes

**Quality:**

2

**Strengths And Weaknesses:**

Strengths:
1. This paper addresses an interesting and challeging task, allowing seamless integration with various assets like
customizable 3D hair, eyeglasses, jewelry, etc.
2. The method introduces a 2d head prior by training diffusion conditioned on skin masks, hair masks , ears and accessory masks, a 3D head mesh rendered in 2D and background information.
3. The authors conduct experiments with both qualitative and quantitative comparisons to show the superiority of proposed method

Weakness:
1.  The proposed method heavily rely on existing components to do straightforward intergration without algorithmic innovation. The authors should provide a clearer explanation of the necessity and unique role of each included condition and module.
2.  It is unclear how well the method generalizes across gender, hairstyles, and ethnicities, large expressions and whether failure cases (e.g., severe occlusion, long hair with background mixing) are handled. How about the performance on integration with various assets like customizable 3D hair, eyeglasses, jewelry, etc.
3. How well does the proposed method handle reconstruction of bald heads when given a reference image? Since the method is titled “HairFree” and emphasizes decoupling accessories and hair, it would be important to know whether the model faithfully reproduces clean scalp geometry and texture in the absence of hair.

---

> ### Author Rebuttal · Authors · 2025-07-31
>
> Thank you for the constructive feedback which we are happy to incorporate in the revised paper. Please find detailed responses and clarifications in the following paragraphs.
>
>
> **(1.1) Explanation of the necessity and unique role of the conditioning signals:**
>
>
> The overall goal of the proposed conditioning strategy is to expose the model to both desired and undesired features during training. This enables the preservation of desired elements and the removal of unwanted ones at inference time. We will add a detailed description of the conditionings to the paper and will show the effect of removing each condition signal in an ablation study (thank you for the suggestion).
>
>
> *- Skin and face mask:*
> During training, the diffusion model is conditioned to learn the appearance of skin and facial regions where specified. At inference time, high-precision 3D skin and face masks are obtained using the FLAME model.
>
> *- Hair mask:*
> The model is trained to recognize and predict hair regions when provided. However, no hair mask is applied during inference. When combined with text prompts indicating a bald head, this approach enables the removal of hair and the generation of clean skin. Without this, hair would be baked into the skin texture.
>
> *- Ear mask:*
> Ear masks are used both during training and inference. During training, masks are generated using an off-the-shelf face parsing model. At inference, accurate 3D ear masks are obtained from FLAME. This ensures consistency between generated and FLAME ears, improving texture fidelity.
>
> *- Accessory mask:*
> Similar to hair masks, accessory masks (e.g., eyeglasses, earrings, clothing) are used during training to guide the model’s learning. These are omitted during inference, allowing for the removal of such accessories and the synthesis of unobstructed skin regions.
>
> *- Rendered 3D head mesh:*
> Conditioning on a rendered 3D head mesh guides the model to generate images consistent with the target geometry and ensures accurate pixel-to-UV space mapping. Training meshes are predicted using an off-the-shelf model, while inference-time meshes can be explicitly defined for texturing. Appendix Figures 21 and 22 (the “LDM Inpaint” row) demonstrate the impact of removing the 3D head mesh conditioning, which leads to reduced geometric consistency and poorer UV mapping.
>
> *- RGB background:*
> Without conditioning on the background, generated images often include undesired elements (e.g., hands, microphones, miscellaneous artifacts) or unnatural shadows. At inference time, setting the background to a uniform color helps prevent such artifacts.
>
>
> **(1.2) Explanation of the necessity and unique role of the different modules:**
>
>
> The presented texture generation approach consists mainly of two modules:
>
> (i) a compositional, 3D-controllable prior which can be used for inpainting in image space, and
>
> (ii) a progressive texture merging/fusion module.
>
> Both components are key contributions of our method to produce bald human textures that can be combined with other 3D assets. The progressive texture generation (ii) iteratively generates a complete texture, by leveraging the prior (i) using the conditionings explained above. The conditioning allows for a good alignment between the 3D model and the generated images which is important to project the colors to the texture space of the FLAME model accurately. The iterative inpainting in image space guarantees that the resulting texture will be seam-less (as the generated images will not have seams) - this is in contrast to methods that do inpainting in texture space. Additionally, Figure 8 shows why inpainting in image space using (i) is important, while Figure 6(A) shows why (ii) is important.
>
>
> **(2) Generalization of the method:**
>
>
> The goal of our method is a text-based generation of bald human head textures for a given 3D FLAME mesh. Based on this mesh and the generated texture, additional assets can be added by the user. For generating the texture, we condition our approach on the given 3D FLAME mesh with a neutral expression.
> In the paper, we show that our method is able to generate textures with a large variation of different attributes and that it also generalizes to the non-photorealistic setting.
> In the photorealistic setting, Figure 4 presents generated results across a range of visual attributes, including gender, age, and ethnicity. Additional examples highlighting diversity in skin tone and nationality are provided in the Appendix (Figures 9 and 10). Figure 4 also includes several celebrity cases, while Appendix Figure 23 offers an extended collection of 360-degree texture maps for various celebrities.
> In the non-photorealistic setting, representative results are shown in Figure 5, with further diverse examples included in the Appendix (Figures 11-20).
>
>
> **(3) Usage of the proposed method for reconstruction:**
>
>
> The proposed method is a generative model that based on a text prompt generates a complete texture of a human head. However, we believe that some of our contributions can also benefit the reconstruction given a reference image. Specifically, our 2D diffusion prior is capable of removing hair from a reference image, enabling it to serve as a bald proxy as shown in Figures 21 and 22 in the Appendix. These generated images could be used as input to a face reconstruction method.
>
>
> **(4) Implementation and training details:**
>
>
> Our method is based on Stable Diffusion 2.1 LDM, which we fine-tune using ControlNet. At inference time, we use the corresponding inpainting prior from Stability AI, both of which are publicly available. The model is fine-tuned for 1500 GPU-hours on a single H100 GPU (see line 144). As mentioned in the submission, we will release the code, pre-trained model, and a dataset of generated samples.
>
>
> Thank you again for your time and constructive feedback. We hope our responses have clarified the key points, and we would be glad to provide further details if needed.

---

> > ### Comment · Reviewer_9xQ3 · 2025-08-05
> >
> > Thanks for the response. Most of my concerns have been addressed. My remaining concern is the necessity of incorporating various signals. I believe there is redundant information among the skin and face mask, as well as the rendered 3D head mesh. I appreciate the systematic design for tackling the human head texturing task.

---

> > > ### Author Response · Authors · 2025-08-06
> > >
> > > Thank you for the follow-up! Here, we briefly clarify the necessity of each conditioning signal for our task to generate a full 360° texture map from text, given a 3D head mesh.
> > >
> > > **Why is conditioning on the rendered 3D mesh necessary?**
> > >
> > > The rendered mesh provides essential spatial guidance (especially the head pose), enabling consistent projection into UV space. The spatial alignment is crucial for texturing, as we use the known projection from the rendered 3D mesh to establish a mapping of the generated pixels to the UV texture space. This precise spatial alignment can not be derived from detected binary semantic maps.
> > >
> > > **Why are semantic maps necessary?**
> > >
> > > The face & skin mask covers all facial regions including eyes, nose, lips, etc, teaching the model what skin and facial features look like (see “Devised Input Conditioning” in System Figure 2). The hair mask is used during training only, helping the model learn hair locations so it can remove hair at inference (e.g., for bald heads). Ear masks are used during both training and inference; at inference, we use precise 3D ear masks from FLAME for accurate, mesh-aligned ear synthesis. Accessory masks allow the model to learn to recognize and remove the accessories by omitting the mask at test time.
> > > Note that during training time those semantic masks can not be derived from the 3D mesh.
> > >
> > > We will include a visual ablation study related to semantic conditioning signals in the revised version.
> > >
> > > Thank you again for your thoughtful question and engagement.

---

### Official Review · Reviewer_h4Gf · 2025-06-23

**Clarity:** 2
**Significance:** 3
**Originality:** 3
**Rating:** 5
**Confidence:** 5

**Summary:**

This paper introduces a method for text-driven bald head texture synthesis based on 3D Morphable Models (3DMM). The approach proceeds in several steps:

1. Fine-tune a Latent Diffusion Model (LDM) with ControlNet on FFHQ and CelebA-HQ datasets, conditioned on parsing maps, foreground masks, and 2D renderings of 3DMM.
2. Train a replacement LDM conditioned on inpainting masks.
3. Generate a frontal head image using LDM and ControlNet, conditioned on the parsing map rendered from 3DMM (only skin and ears), foreground mask, and image, then unwrap the image into an atlas.
4. For each view, render a textured 3DMM and perform inpainting using the LDM, then unwrap the pixels into the final texture map.

Overall, the paper provides a baseline for a valuable yet underexplored task. While there are minor mistakes or unclear parts in the paper, they do not detract from the significance and novelty of the proposed work.

**Questions:**

See the weakness above.

**Ethical Concerns:**

["NO or VERY MINOR ethics concerns only"]

**Final Justification:**

I appreciate the authors' efforts and detailed response. All my previous concerns are addressed. Although the metrics issues regarding the insufficient number of bald head data for a comprehensive evaluation remain unsolved, this paper proposed a novel solution for an interesting research problem. I think this minor concern can be solved in the revision with more visualization. In summary, I would like to increase my score and suggest an acceptance recommendation.

**Limitations:**

Yes

**Quality:**

3

**Strengths And Weaknesses:**

### Strengths

1. The method establishes a baseline for the task of text-driven bald head texture synthesis, which has practical applications but has not been extensively studied.
2. The idea is clever. It leverages priors from FFHQ and CelebA-HQ datasets to fine-tune ControlNet conditioned on parsing maps, and uses 3DMM-rendered parsing maps with only skin and ears to guide novel face generation.
3. The method shows good generalizability; it can generate non-human skin textures based on textual prompts.
4. The pipeline is intuitive and reasonable. The experimental results are sufficient, including comparisons and diversified experiments that demonstrate the method's effectiveness and efficiency.

### Weaknesses
1. Miscellaneous Issues:
    1. In Figure 2(A), why is the label for the last row "RGB"? Please clarify.
    2. In Figure 2(B), does $\beta$ denote ControlNet strength? Also, what does $\epsilon$ represent?
    3. In Figures 21 and 22, the legend labels appear to be flipped.
    4. In Figure 6(B), the figure lacks a legend indicating that the first row corresponds to higher CS values, and the second to lower ones.
2. In line 161: "we render the mesh from 14 viewpoints by rotating a virtual camera around the head" — Why was 14 chosen? An ablation study on the number of viewpoints would strengthen the justification.
3. 1-to-many issues:
    1. In Equation (9), is it possible for two pixels to map to the same UV texel? If so, how is this ambiguity resolved?
    2. In Equation (11), can a single texel receive values from four different adjacent grids? How is this handled?
4. In line 204: What is "ControlNet Strength (CS)"? This term does not appear in the original ControlNet paper and should be clearly defined.
5. In lines 204–205: "By reducing CS, we can generate more diverse examples, enabling the synthesis of novel head views that do not exist in the training data (Table 2)." However, Table 2 shows that all metrics degrade as CS decreases. Therefore, the claim about improved diversity lacks sufficient evidence.
6. The content in Section 5.1 "Quantitative Evaluation" and Section 5.2 "Ablation - ControlNet Strength (CS)" is somewhat redundant. Moreover, the conclusion of Ablation - ControlNet Strength (CS) to be more relevant to Section 5.1.
7. In Section 5.2 "Comparison with Inpainting Methods": What does Content-Aware Fill (CAF) refer to? Please provide more technical details or cite relevant literature.

---

> ### Author Rebuttal · Authors · 2025-07-31
>
> We are thankful for the positive and constructive feedback! In the following, we will clarify the questions raised under ‘weaknesses’ and we will incorporate the suggestions and answers into the revised paper.
>
>
> **(1) Miscellaneous issues:**
>
> - *In Figure 2(A), why is the label for the last row "RGB"? Please clarify.*
>
>    - R, G, B refers to the three usual RGB channels. We will rename it to Channel 1, Channel 2, and Channel 3 for clarity.
>
> - *In Figure 2(B), does $\beta$ denote ControlNet strength? Also, what does $\epsilon$ represent?*
>
>    - Yes, $\beta$ is ControlNet strength. $\epsilon$ is a small neural network through which the input condition downsamping is learned (Zhang et al. [68]).
>
> - *In Figures 21 and 22, the legend labels appear to be flipped.*
>
>   - Thank you for noticing this, we will put them back on top!
>
>  - *In Figure 6(B), the figure lacks a legend indicating that the first row corresponds to higher CS values, and the second to lower ones.*
>
>    - Yes, we will add the legend for clarity, thank you!
>
>
> **(2) An ablation study on the number of viewpoints would strengthen the justification.**
>
>
> We empirically found that reducing the number of side-views may lead to the Janus artifact at the back of the head (similar to Figure 6(A)). We will add examples of this when fewer views are used to the revised version.
>
>
> **(3) One-to-many issues:**
>
>
> We use a splatting-based approach to set the texel values given a specific pixel and corresponding UV coordinate. Thus, two pixels could be mapped to the same UV texels. Here, a distance based blending scheme could be used, but we found that assigning one of the values without blending yields good results.
>
>
> **(4) What is "ControlNet Strength (CS)"?**
>
>
> ControlNet is a copy of the original LDM U-Net that processes the control (conditioning) image separately. Its outputs are then added to the original U-Net’s outputs at corresponding blocks during the denoising process. This allows the model to integrate external structural guidance (like pose, edges, or depth) without modifying the base U-Net. The degree to which this control input influences the final image can be adjusted using the ControlNet strength parameter. We will describe this in more detail in the revised paper. Specifically, let $F_i$ be the output from the original LDM U-Net at block $i$, $C_i$ be the corresponding ControlNet feature map at the same stage, and $\beta$ be the ControlNet strength parameter. Then, the combined feature at each block during the denoising process is computed as: $F_i^{\text{new}} = F_i + \beta \cdot C_i$
>
>
> **(5) In lines 204–205: "By reducing CS, we can generate more diverse examples, enabling the synthesis of novel head views that do not exist in the training data (Table 2)." However, Table 2 shows that all metrics degrade as CS decreases. Therefore, the claim about improved diversity lacks sufficient evidence.**
>
>
> Table 2 shows that all metrics degrade as CS decreases, considering the data distribution that the model was fine-tuned on. To remember, this data contains mostly frontal and some side-views of photo-realistic human faces with hair. In our case, we want to generate samples that are different from the original data distribution in three aspects (1) no hair (bald examples), (2) various camera views including top and back-views), (3) non-photorealistic (including non-human) examples. Therefore, the goal is for the metrics to decrease as ControlNet strength decreases which is what is shown in the table (because data distribution that we want to generate differs from the one that is present in data). In other words, metrics degrading as CS reduces (there is a trend, lower CS leads to worse metrics’ scores) signifies that the distribution of generated data shifts more and more from the original distribution, which is what we want.
>
>
> **(6) The content in Section 5.1 "Quantitative Evaluation" and Section 5.2 "Ablation - ControlNet Strength (CS)" is somewhat redundant. Moreover, the conclusion of Ablation - ControlNet Strength (CS) to be more relevant to Section 5.1.**
>
>
> In Section 5.1 we were referring to quantitative evaluation of the CS parameter, while in section 5.2 this was shown qualitatively. We will merge those two in Section 5.1 in the revised version.
>
>
> **(7) In Section 5.2 "Comparison with Inpainting Methods": What does Content-Aware Fill (CAF) refer to? Please provide more technical details or cite relevant literature.**
>
>
> CAF is an “industrial standard” for inpainting in professional tools like PhotoShop.
>
> Thank you again for your time and constructive feedback. We hope our responses have clarified the key points, and we would be glad to provide further details if needed.

---

> ### Comment · Reviewer_h4Gf · 2025-08-04
>
> Thank you for the rebuttal; most of my concerns have been addressed. However, I still have some questions that I want to discuss with you.
>
> Considering the response (3), does the distance-based blending scheme used follow the original strategy in 3D Gaussian splatting? Since there has been research showing that the 3D Gaussian splatting presents the view-dependent appearance issues that may introduce flicker artifacts when the view camera shifts around certain angles. Did this phenomenon occur in your experiment?
>
> For response (5), it is reasonable that reducing CS leads to a degradation of quantitative results regarding the data distribution inconsistency. However, it cannot be said that this observation exactly follows what the author thought. Since inferior synthesized results can also make the metrics worse, could the authors provide more evidence (such as visualizations or more appropriate metrics) to illustrate this finding? Or just found out some examples where the bald texture of the back and side-head is available but not used for training, and calculate the metrics on this constructed evaluation set?

---

> > ### Author Response · Authors · 2025-08-05
> >
> > Thank you for your response and follow-up questions. Please find our clarifications below:
> >
> >
> > **(3) View-dependent appearance issues in 3D Gaussian splatting:**
> >
> >
> > In 3D Gaussian Splatting, geometry is represented as a set of volumetric primitives (3D Gaussians / ellipsoids), which can introduce various rendering artifacts:
> >
> > - *Depth-fighting artifacts*, caused by incorrect accumulation order. These effects resemble z-fighting and are addressed in methods like Stop the Pop (Radl et al.).
> >
> >
> > - *Floater artifacts*, which occur due to overfitting or insufficient view coverage. In these cases, additional Gaussians are placed in front of the actual surface to reproduce the input image, resulting in a lack of a clear, well-defined surface.
> >
> >
> > In contrast, our method uses a clearly defined and solid surface geometry, without any transparency. The splatting process maps the observed surface color to its corresponding location in UV texture space, by projecting from image space to texture coordinates.
> > Because we operate on a well-defined surface and use a view-independent texture, our approach does not suffer from the appearance artifacts common in volumetric splatting methods.
> >
> > **(5) ControlNet strength influence on distribution shift:**
> >
> >
> > The ControlNet strength has a major influence on shifting from one distribution to another. Unfortunately, we don’t have a large dataset of bald heads, where we could directly evaluate the generation of the backside of the head. However, we conducted an experiment on changing the distribution from human faces to cat faces (which is an extreme case of a distribution shift), where we could analyse the effects of the ControlNet Strength.
> >
> > Specifically, we applied our method to the AFHQ-Cat dataset from StarGAN v2 [Choi et al., 2020], a commonly used dataset from a significantly different domain. We measured FID and KID scores at various $\beta$ values:
> >
> > Beta     | FID     | KID
> > ---------|---------|---------
> > 1.0      | 231.32  | 0.2074
> > 0.75     | 228.46  | 0.2035
> > 0.5      | 212.10  | 0.1910
> > **0.25**     |  **39.88**  | **0.0237**
> > 0        |  81.93  | 0.0347
> >
> > For higher values of ControlNet strength (e.g., $\beta \geq 0.5$), the model remains biased toward synthesizing human faces. This prevents effective generation of samples from the cat distribution, resulting in high FID/KID scores and poor visual alignment with the AFHQ domain.
> >
> > At $\beta = 0.25$, the ControlNet guidance is reduced enough to allow the diffusion prior to generate realistic cat faces, while still retaining enough conditioning to preserve global head structure. This balance aligns well with the AFHQ-Cat distribution, as reflected in the significantly improved scores.
> >
> > At $\beta = 0$, although the model continues to produce cat faces, the samples are often zoomed-in facial crops rather than full cat heads, deviating from the AFHQ distribution. In contrast, at $\beta = 0.25$, structural guidance helps preserve head framing consistent with the dataset.
> >
> > We will add visualizations to the revised paper, and discuss the importance of the ControlNet strength to achieve this distribution shift.
> >
> > **References:**
> >
> > *Yunjey Choi, Youngjung Uh, Jaejun Yoo, Jung-Woo Ha. StarGAN v2: Diverse Image Synthesis for Multiple Domains. In CVPR, 2020.*
> >
> > *Lukas Radl, Michael Steiner, Mathias Parger, Alexander Weinrauch, Bernhard Kerbl, Markus Steinberger. StopThePop: Sorted Gaussian Splatting for View-Consistent Real-time Rendering. In ACM Transactions on Graphics (TOG), 2024.*
> >
> >
> > Please feel free to reach out if you have any further questions.

---

> > > ### Comment · Reviewer_h4Gf · 2025-08-05
> > >
> > > I appreciate the authors' efforts and detailed response. All my previous concerns are addressed. Although the metrics issues regarding the insufficient number of bald head data for a comprehensive evaluation remain unsolved, this paper proposed a novel solution for an interesting research problem. I think this minor concern can be solved in the revision with more visualization. In summary, I would like to increase my score and suggest an acceptance recommendation. Thanks again for the authors' works. Goodluck!

---

> > > > ### Author Response · Authors · 2025-08-06
> > > >
> > > > Thank you!

---

### Decision · Program_Chairs · 2025-09-17

**Decision:**

Accept (poster)

**Comment:**

The paper has initially received thorough reviews. The reviewers also discussed the rebuttal in depth.

The rebuttal has helped address a lot of concerns, and the reviewers agreed to positively score the paper.

In the current form, we recommend Accept. Congratulations!
We encourage the authors to take into account the reviewers' comments for the final version (see below).

* Although the metrics issues regarding the insufficient number of bald head data for a comprehensive evaluation remain unsolved, this paper proposed a novel solution for an interesting research problem. A Reviewer thinks this minor concern can be solved in the revision with more visualization.

* One Reviewer remaining concern is the necessity of incorporating various signals. They believe there is redundant information among the skin and face mask, as well as the rendered 3D head mesh. While they appreciate the systematic design for tackling the human head texturing task, the composition of existing techniques still lacks strong methodological novelty. The authors are encouraged to release the code to make the design choices in each part clearly.

* Two Reviewers find that technical contribution is somewhat limited. We encourage the Authors to further clarify their contributions.